# DDR2 controls breast tumor stiffness and metastasis by regulating integrin mediated mechanotransduction in CAFs

Samantha VH Bayer[1,2,3], Whitney R Grither[1,3,4], Audrey Brenot[1,3], Priscilla Y Hwang[1,3], Craig E Barcus[1,3], Melanie Ernst[1,4], Patrick Pence[1], Christopher Walter[5], Amit Pathak[5], Gregory D Longmore[1,2,3]*

[1]ICCE Institute, Washington University, St Louis, United States; [2]Department of Cell Biology and Physiology, Washington University, St Louis, United States; [3]Department of Medicine, Washington University, St Louis, United States; [4]Department of Biochemistry, Washington University, St Louis, United States; [5]Department of Mechanical Engineering, Washington University, St Louis, United States

**Abstract** Biomechanical changes in the tumor microenvironment influence tumor progression and metastases. Collagen content and fiber organization within the tumor stroma are major contributors to biomechanical changes (e., tumor stiffness) and correlated with tumor aggressiveness and outcome. What signals and in what cells control collagen organization within the tumors, and how, is not fully understood. We show in mouse breast tumors that the action of the collagen receptor DDR2 in CAFs controls tumor stiffness by reorganizing collagen fibers specifically at the tumor-stromal boundary. These changes were associated with lung metastases. The action of DDR2 in mouse and human CAFs, and tumors in vivo, was found to influence mechanotransduction by controlling full collagen-binding integrin activation via Rap1-mediated Talin1 and Kindlin2 recruitment. The action of DDR2 in tumor CAFs is thus critical for remodeling collagen fibers at the tumor-stromal boundary to generate a physically permissive tumor microenvironment for tumor cell invasion and metastases.
DOI: https://doi.org/10.7554/eLife.45508.001

*For correspondence:
glongmore@wustl.edu

**Competing interests:** The authors declare that no competing interests exist.

## Introduction

In addition to inherited and acquired mutations in tumor cells, cancer development, progression, invasion, metastases, and responses to therapy require a permissive tumor microenvironment. The tumor microenvironment consists of activated cells, (e.g., immune, cancer associated fibroblasts (CAFs), endothelial cells) deposited growth factors and cytokines, and extracellular matrix (ECM) and these all differ in number, composition and function from the tissue environment surrounding normal epithelia. For example, biophysical changes resulting from increased matrix deposition and remodeling result in mechanical perturbations, such as increased tissue stiffness. Increased tumor stromal stiffness can influence cell differentiation (transformation), cell proliferation, cell invasion and migration, and biochemical signaling by embedded growth factors (*Paszek et al., 2005*) (*Provenzano et al., 2008*) (*Schedin and Keely, 2011*). Increased and altered collagen fiber production as well as ECM collagen fiber remodeling, in particular, are major contributors to the altered stiffness of tumors, especially in pancreatic and breast tumors.

In breast tumors, stromal collagen fiber characteristics differ significantly from their normal tissue counterparts in amount, composition, architecture, and function (*Bonnans et al., 2014*). As a result,

aggressive breast tumors are stiffer than corresponding normal tissues and this feature can be used to detect, stage, and prognosticate, as well as limit treatment efficacy. Collagen fibers are the most abundant protein in the extracellular matrix (ECM) of breast tumors and women with dense breasts, due in part to increased collagen deposition, have an increased risk for developing breast cancer and when they do their cancers are more aggressive (*Aiello, 2005*). Collagen fiber alignment or orientation at the tumor-stromal boundary, collagen fiber thickness, length, and architecture all affect the mechanical properties, or stiffness, of primary breast tumors and directly impact tumor invasion and metastatic progression (*Conklin et al., 2011*). Indeed, a clinical tumor associated collagen signature (TACS) has been developed and correlates to aggressiveness of breast tumors. Collagen fibers that are short, curly, and thin are correlated with benign or less aggressive tumors while thicker fibers that align perpendicular to the tumor-stromal interface are indicative of aggressive tumors (*Provenzano et al., 2006*). Despite its clinical importance how tumor associated collagen fiber organization is generated is not fully appreciated.

Accumulated data indicates that control of ECM matrix in tumors is likely multifactorial, but the major effector cell within the tumor microenvironment most responsible for ECM matrix production and arguably remodeling is the cancer associated fibroblast (CAF) (*Bhowmick et al., 2004*). In addition to producing ECM components, CAFs remodel the tumor stroma through: (1) secretion of ECM remodeling enzymes such as collagen cross-linking enzymes (e.g., lysyl-oxidases) and proteases (e.g., MMPs and collagenases) and (2) force mediated matrix remodeling. Other cell types within the tumor stroma such as immune cells and tumor cells can also impact ECM remodeling either directly (e.g., secretion of MMPs, LOX, or collagenases) or indirectly (e.g, secretion of growth factors and cytokines that activate CAFs) (*Acerbi et al., 2015*). Therefore, understanding CAF cell-intrinsic regulation of ECM production and collagen fiber remodeling during cancer progression and metastasis is an important consideration not only for multiple cancers, but also other fibrotic diseases, and treatment strategies for both.

In this regard, genetically engineered mouse tumor models (GEMM) and syngeneic orthotopic transplant models have been informative. In both these models the action of a cell surface fibrillar collagen receptor DDR2 in tumor cells and tumor stromal cells has been shown to regulate breast cancer metastasis (*Zhang et al., 2013*) (*Zhang et al., 2014*) (*Corsa et al., 2016*) (*Gonzalez et al., 2017*). But, in which cells within the tumor environment the action of DDR2 is important and how is not known. Normal breast epithelium does not express DDR2, however, in invasive human breast tumor cells DDR2 expression is induced in 50–70% of cases and is significantly correlated with poor outcomes, particularly in aggressive TNBCs (*Zhang et al., 2013*) (*Toy et al., 2015*). DDR2 is also an unusual receptor tyrosine kinase (RTK) (*Fu et al., 2013*) (*Leitinger, 2014*). In contrast to other RTKs its ligand is a structural protein (fibrillar collagen) rather than a growth factor or cytokine, and its activation and inactivation kinetics are slow for reasons not understood. Despite its ligand, DDR2 is unlikely to be a significant adhesive receptor alone, but it has been suggested that its action may affect collagen binding Integrin affinity (*Xu et al., 2012*). The molecular mechanism(s) and functional significance of this observation has not been determined.

The other major collagen receptor in tumor cells and tumor stromal cells are the collagen binding Integrins. Genetic and pharmacologic studies have shown that Integrins are also critical for cancer development and metastasis (*Bianconi et al., 2016*) (*Hamidi and Ivaska, 2018*). In GEMMs deletion of *Ddr2* appears to have a greater impact on breast cancer metastasis than does either α1 or α2 Integrin deletion (two α chains of collagen binding integrins), while β1 Integrin plays a critical role in tumor initiation and maintenance (*Lahlou and Muller, 2011*) (*Ramirez et al., 2011*) (*White et al., 2004*). Integrin and DDR2 have distinct, non-overlapping binding sites within fibrillar collagens and DDR2 can be activated by collagen in the absence of integrins (*Vogel et al., 1997*). In contrast to DDR2, integrins are bona fide adhesion molecules as well as signaling receptors. A major function of integrins is in environmental mechanosensing and mechanotransducing (*Sun et al., 2016*), and thus, are sensitive and responsive to changes in the mechanical properties of the cellular environment.

Here we show that genetic deletion of the *Ddr2* gene in breast tumor CAFs, without altering DDR2 expression in tumor cells, impacts their mechanotransduction properties. It does so by activating Rap1 with subsequent activation and, or recruitment of Talin1 and Kindlin2 to cell surface β1 Integrin. As a result, DDR2 is selectively required for full activation of collagen binding Integrins in CAFs, as fibronectin activated Integrins are normal. In vivo, breast tumors in which *Ddr2* is deleted in CAFs are less stiff, have an altered collagen fiber organization particularly at the tumor-stromal

boundary, and decreased β1 Integrin activity. These changes are associated with decreased lung metastasis. These data indicate that the action of DDR2 is an important regulator of mechanotransduction in breast tumor CAFs, critical for full activation of collagen-binding Integrins and the formation of a metastasis permissive biophysical tumor environment.

## Results

### The action of DDR2 within stromal cells of the primary tumor site, as opposed to a metastatic site, impact breast cancer lung metastases

DDR2 expression in stromal cells of primary breast tumors as well as in stromal cells of lung metastases is increased, and reciprocal orthotopic syngeneic breast tumor transplant experiments have revealed that the action of DDR2 within stromal cells of the recipient host regulate breast cancer lung metastases (*Corsa et al., 2016*). The anatomic site of action (primary tumor or metastatic site or both), the particular stromal cell type(s) responsible, and the cellular molecular mechanisms involved are not known, however. To determine whether the action of DDR2 in metastatic sites was critical, we determined the extent of lung colonization by wild type primary MMTV-PyMT breast tumor cells following tail vein injection of into syngeneic $Ddr2^{+/+}$ or ubiquitous $Ddr2^{-/-}$ hosts. There was no difference in the number, size, or total volume of lung tumors between either recipient (*Figure 1A*). In contrast, when WT (DDR2 +ve) breast tumor cells were transplanted into syngeneic $Ddr2^{+/+}$ or ubiquitous $Ddr2^{-/-}$ hosts there was a decrease in primary tumor volume as well as significantly decreased number of lung metastases (*Figure 1B and C*). Viewed together, these results suggested that the action of DDR2 in the stromal cells within the primary tumor environment, as opposed to the action of DDR2 in stromal cells at lung metastatic site, was important in influencing metastasis.

### DDR2 affects mechanotransduction functions of CAFs

In primary breast tumors of ubiquitous $Ddr2^{-/-}$; MMTV-PyMT mice the collagen fiber organization surrounding tumor nodules is altered compared with $Ddr2^{+/+}$; MMTV-PyMT breast tumors (*Corsa et al., 2016*). Collagen fiber organization in breast tumors from ubiquitous $Ddr2^{-/-}$ mice are more benign or less supportive of tumor invasion and spread and these mice develop significantly

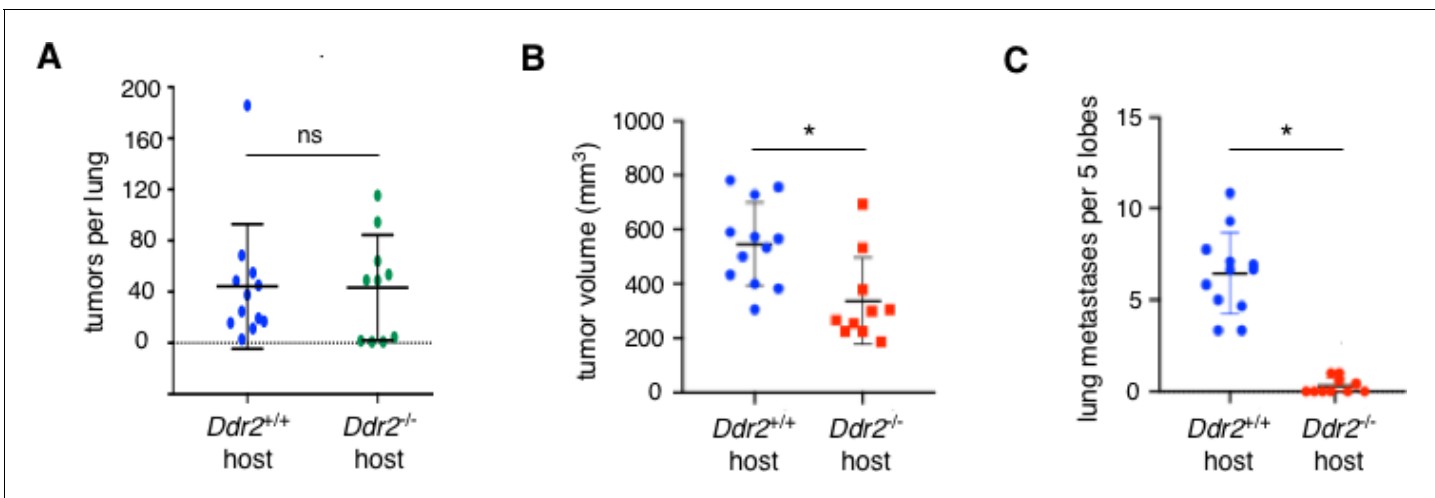

**Figure 1.** The primary effect of DDR2 in stromal cells is at the primary tumor site not lung metastatic site. (**A**) Lung colonization assay. Number of tumors per lung at 2 weeks after tail vein injection of GFP-Luciferase PyMT breast tumor cells ($Ddr2^{+/+}$) into syngeneic FVB/n $Ddr2$ WT ($Ddr2^{+/+}$) mice (11 mice; n = 11) or ubiquitous $Ddr2$ null ($Ddr2^{-/-}$) mice (10 mice; n = 10). (**B and C**) Orthotopic transplant metastasis assay. DDR2 +ve 4T1 breast tumor cells were transplanted into the breast of syngeneic Balb/c $Ddr2^{+/+}$ (12 mice; n = 12) or ubiquitous $Ddr2^{-/-}$ (10 mice; n = 10) recipients. Primary tumor volume (**B**) and number of lung metastases (**C**) were scored at 3 weeks after transplant. In all experiments, tumors were scored and enumerated histologically. Statistics were one-way ANOVA with Tukey's post hoc test. *p<0.05, ns - no significant difference.

DOI: https://doi.org/10.7554/eLife.45508.002

reduced lung metastases. Since CAFs are the predominant tumor stromal cell within the primary tumor environment that controls ECM production and arguably ECM remodeling (*Bhowmick et al., 2004*) and the *Ddr2* gene and DDR2 protein expression is significantly upregulated in breast tumor CAFs during cancer progression (*Corsa et al., 2016*) (*Gonzalez et al., 2017*), we asked whether the action of DDR2 in breast tumor CAFs impacted CAF cellular functions that facilitate tumor progression and metastasis.

We isolated primary mouse CAFs (mCAFs) from *Ddr2*fl/fl; MMTV-PyMT (*Ddr2*+/+), Actin-cre; *Ddr2*fl/fl; MMTV-PyMT (*Ddr2*-/-), and S100a4-cre (also called FSP1cre); *Ddr2*fl/fl; MMTV-PyMT (*Ddr2*-/-) mice as well as fibroblasts from normal mouse breast (*Figure 2—figure supplement 1A*). In addition, *Ddr2* expression was depleted in an immortalized human breast tumor CAF cell lines (hCAFs) using shRNA expressing lentiviruses (*Zhang et al., 2016*) (*Figure 2—figure supplement 1B*). In 2D cultures, WT CAFs produce a linear, ordered collagen fibrillar matrix while normal fibroblasts produce a more disorganized collagen matrix (*Corsa et al., 2016*). In mCAFs lacking DDR2 the collagen matrix produced in culture was more like the matrix produced by normal fibroblasts: disorganized (*Figure 2—figure supplement 1C*). Re-expression of WT DDR2 into *Ddr2*-/- mCAF reverted matrix production to that typical of WT mCAFs: ordered and linear (*Figure 2—figure supplement 1C*).

CAFs adhere and spread on collagen I-coated plates and when embedded in 3D collagen I gels, contract collagen fibers. We measured and contrasted mCAF adhesion and spreading on 2D collagen I-coated plates. *Ddr2*-/- CAFs exhibited decreased cell spreading that could be rescued by re-expression of WT DDR2, but not tyrosine kinase-inactive DDR2 <K608E> (*Figure 2A*). When *Ddr2*-/- mCAFs and *Ddr2* depleted hCAFs were embedded in 3D collagen I gels, gel contraction was inhibited compared to WT CAFs (*Figure 2—figure supplement 1D*).

A possible explanation for these phenotypes could be altered focal adhesion (FA) formation and, or function. To test this possibility, we first measured and contrasted FA size in various mouse and human CAFs when plated on collagen I coated plates for 30 or 60 min or overnight. For all time points examined both human CAFs depleted of *Ddr2* and mouse CAFs deleted of *Ddr2* exhibited smaller FAs (*Figure 2B and C* ; *Figure 2—figure supplement 2A*). When CAFs were plated on Fibronectin there was no difference in the size of focal adhesions (*Figure 2—figure supplement 2B*). Despite the difference in focal adhesion size, there was no significant difference in the number of FAs between cells plated on collagen I (*Figure 2—figure supplement 2C*). Re-expression of WT DDR2, but not collagen binding defective DDR2 <W52A>, in *Ddr2* depleted hCAFs rescued this defect (*Figure 2D*).

To determine if there were functional consequences as a result of these changes in CAF cell biology, we measured the traction force generated by various CAFs on collagen I coated hydrogels. *Ddr2*-depleted hCAFs exhibited dramatically reduced traction force, and this could be rescued by re-expressing WT DDR2, but not collagen binding defective DDR2 <W52A> (*Figure 2E*). Consistent with these changes in contractile properties, both *Ddr2*-/- mCAFs and hCAFs depleted of *Ddr2* exhibited diminished pMLC activity when plated on Collagen I (*Figure 2—figure supplement 2D*). The defect in pMLC activity could be rescued by re-expression of WT DDR2 but not collagen binding defective DDR2 <W52A> (*Figure 2—figure supplement 2D*). These results were not a result of a differential level of rescue isoform expression as all were expressed in equivalent amounts as determined by total cellular DDR2-YFP fluorescence (*Figure 2—figure supplement 3A*) and Western blot (*Figure 2—figure supplement 3B*). Furthermore, all rescue isoforms were expressed on the cell surface as determined by cross-linking with the cell-impermeant reagent BS[3] (*Figure 2—figure supplement 3C*).

In sum, these experiments indicated that DDR2 deficient mouse and human breast CAFs exhibited altered mechanotransduction activity when plated on or in collagen I.

## DDR2 is required for full activation of collagen binding β1 integrin in CAFs

DDR2 alone is not a strong adhesive receptor (*Xu et al., 2012*). Integrins, however, are considered major cellular receptors that sense and respond to changes in ECM mechanical properties (*Sun et al., 2016*). Therefore, we asked whether the mechanotransducing effects of DDR2 in CAFs could involve regulation of Integrin activity. All collagen binding Integrins contain variable α chains and a common β1 chain. Human CAFs were plated on collagen I, fixed and stained with the 9EG7 antibody that detects active β1 Integrin in both human and mouse cells (*Bazzoni et al., 1995*;

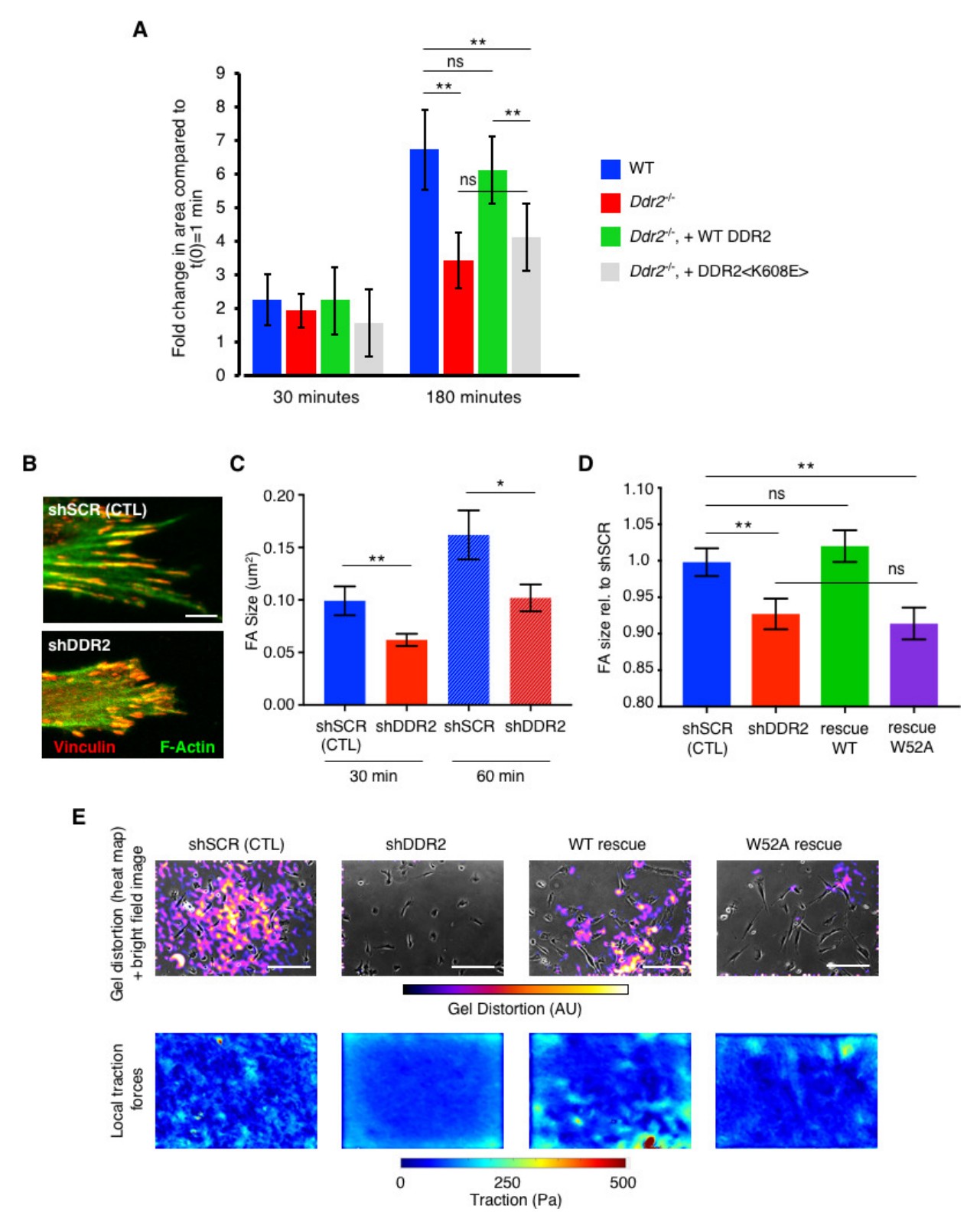

**Figure 2.** DDR2 influences mechanotransduction by cancer associated fibroblasts. (A) Cell Spreading Assay. Mouse breast CAFs were added to collagen I coated plates for 30 or 180 min. Blue columns WT CAFs; red columns *Ddr2*-/- CAFs; green columns *Ddr2*-/- CAFs rescued with WT DDR2; gray columns *Ddr2*-/- CAFs rescued with tyrosine kinase dead mutant DDR2 <K608E>. Cell area was determined from phase contrast image in movies. Time = 0 was the area of each cell 10 min after plating. All values are normalized to t = 0 for each cell type which was arbitrarily set to equal 1. At least

*Figure 2 continued on next page*

*Figure 2 continued*

100 cells for each genotype were analyzed. (**B**) Representative confocal images of focal adhesions (as detected by Vinculin (red) and F-Actin (green) immunofluorescence) in WT (shSCR) or *Ddr2*-depleted (shDDR2) hCAFs plated on collagen I coated coverslips overnight. Scale bar = 10 um. (**C**) Quantification of focal adhesion size of indicated hCAFs plated on collagen I coated coverslips after 30 or 60 min. (**D**) Quantification of focal adhesion size of indicated hCAFs after plating on collagen I for 18 hr. For all focal adhesion measurements in each cell type and time point, >600 total focal adhesions from 20 different cells were measured. (**E**) Traction forces generated by indicated hCAF plated on collagen I-coated soft polyacrylamide gels embedded with fluorescent beads. Top row- heat map of bead displacement field overlaid on bright field image of cells; bottom row- field of calculated forces. Scale bars = 100 um. In all panels, statistics were one-way ANOVA with Tukey's post hoc test. Unless otherwise noted, *p<0.05, **p<0.01, ***p<0.001, ns no significant difference.

DOI: https://doi.org/10.7554/eLife.45508.003

The following figure supplements are available for figure 2:

**Figure supplement 1.** DDR2 influences mechanotransduction by cancer associated fibroblasts supplement 1.

DOI: https://doi.org/10.7554/eLife.45508.004

**Figure supplement 2.** DDR2 influences mechanotransduction by cancer associated fibroblasts supplement 2.

DOI: https://doi.org/10.7554/eLife.45508.005

**Figure supplement 3.** DDR2 influences mechanotransduction by cancer associated fibroblasts supplement 3.

DOI: https://doi.org/10.7554/eLife.45508.006

*Su et al., 2016*) or Talin1 and then super-resolution confocal microscopy was performed. The amount of active β1-Integrin and Talin1 at adhesion sites in cellular protrusions was quantified from images. In *Ddr2* depleted hCAFs significantly less active β1 Integrin and Talin1 was present in cell protrusions (*Figure 3A and C*). When active β1 Integrin-Talin1 co-localization in cell protrusions was determined there was significantly less co-localization in *Ddr2* depleted hCAFs compared with control shSCR hCAFs (*Figure 3C*). Like Talin1, Kindlin2 is another cytoplasmic protein recruited to Integrin adhesive sites and important for full Integrin activity (*Calderwood et al., 2013*). Kindlin2 - active β1 Integrin and Talin1-Kindlin2 co-localization in cell protrusion were both significantly decreased in *Ddr2* depleted hCAFs (*Figure 3—figure supplement 1A and B*). This result was not due to decreased expression of β1 Integrin in *Ddr2*-depleted CAFs, as when attached and spread cells were stained with an antibody that detected total β1 Integrin there was no difference in the total amount of β1 Integrin in cell protrusions of WT versus *Ddr2* depleted cells (*Figure 3B* ; *Figure 3—figure supplement 1C and D*). Furthermore, Western blots of *Ddr2* depleted hCAFs and WT hCAFs cell extracts demonstrated that there was no significant difference in the total cellular β1 Integrin and Talin1 levels between cells (*Figure 3D*). Consistent with these imaging results, the amount of β1-Integrin and Talin1 that co-immunoprecipitated in *Ddr2* depleted hCAFs plated on collagen I was significantly reduced (*Figure 3D*).

β1-Integrin chains are also used by some fibronectin binding Integrins, so we asked whether this was a general defect in β1-containing Integrin activation or selective for collagen interacting β1-containing Integrins. When *Ddr2* depleted hCAFs were plated on fibronectin there was no difference in the amount of active β1-Integrin and Talin1 present at adhesion sites within cellular protrusions from what was present in WT hCAFs (*Figure 3E*). Furthermore, when *Ddr2*−/− mCAFs and WT mCAFs were plated on Fibronectin coated plates there was no difference in FA size noted (*Figure 2—figure supplement 2B and C*).

These results indicated that the presence of DDR2 in CAFs influenced full activation of collagen binding Integrins at adhesive sites within cell protrusions by controlling Talin1 and Kindlin2 activation and, or recruitment to Integrin adhesion complexes in cell protrusions.

## DDR2 controls β1 integrin activity via Rap1 mediated Talin1 and Kindlin2 activation and recruitment to collagen binding β1 integrins

Both DDR2 and β1-Integrin localized to adhesive sites within cell protrusions, but calculation of the Pearson's correlation coefficient revealed that, compared to Talin1, DDR2 did not colocalize with β1-Integrin at these sites (*Figure 4A*). Defects in Talin1 and Kindlin2 recruitment to cell surface Integrin complexes in cell protrusions of *Ddr2* depleted hCAFs suggested the possibility that DDR2 might affect 'inside-out' regulation of Integrin activation. A significant regulator of Talin1 activation and recruitment to cell surface Integrins is the cytosolic GTP-binding protein RAP1. When WT control hCAFs were exposed to collagen I, RAP1 was activated (*Figure 4B*). In *Ddr2* depleted hCAFs there

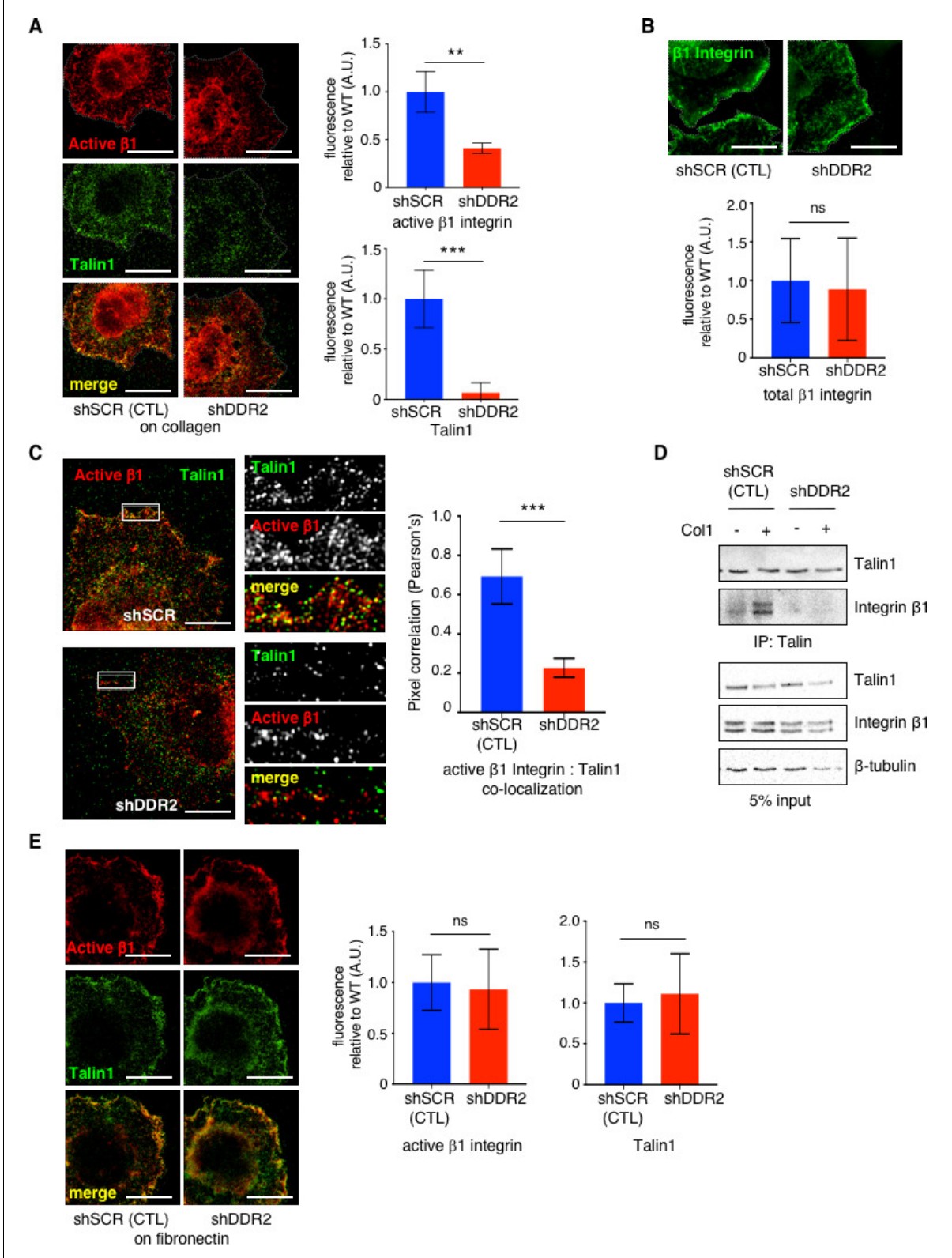

**Figure 3.** DDR2 is required for full activation of collagen binding β1 Integrin in CAFs. (**A**) Representative n-SIM super-resolution microscopy images of control hCAFs (shSCR) or hCAFs depleted of *Ddr2* (shDDR2) showing active integrin β1 (9EG7; red) and Talin1 (green) immunofluorescence after plating cells for 15 min on collagen I coated coverslips. Quantified in graphs on right. Fluorescence of each protein at cell surface of protrusions in WT cells was arbitrarily set = 1, and at least 20 cells for each genotype were analyzed. (**B**) Representative n-SIM super-resolution microscopy images of control

*Figure 3 continued on next page*

Figure 3 continued

hCAFs (shSCR) or hCAFs depleted of *Ddr2* (shDDR2) showing total integrin β1 (green) immunofluorescence after plating cells for 15 min on collagen I coated coverslips. Quantified in graph below. Fluorescence at cell surface of protrusions in WT cells was arbitrarily set = 1 and at least 20 cells were analyzed. (C) Representative n-SIM super-resolution microscopy images of control hCAFs (shSCR) or hCAFs depleted of *Ddr2* (shDDR2) showing active integrin β1 (9EG7; red) and Talin1 (green) immunofluorescence after plating cells for 15 min on collagen I coated coverslips. Images on right are higher resolution images of area within the white box. Quantification of active integrin β1 (9EG7) and Talin1 co-localization (Pearson's coefficient) is presented in graph on right. At least 20 cells of each genotype were analyzed. ***p<0.0004. (D) Coimmunoprecipitation of Talin1 and integrin β1 (upper panel) and 5% input control (lower panel) from control hCAF (shSCR) and hCAF depleted of DDR2 (shDDR2) cells plated on collagen I coated plates (+) or no collagen (-) for 1 hr. This a representative result of 1 of 3 separate experiments. (E) Representative n-SIM super-resolution microscopy images of control hCAF (shSCR) or hCAF depleted of *Ddr2* (shDDR2) plated on fibronectin coated coverslips for 15 min showing active integrin β1 (9EG7, red) or Talin1 (green) immunofluorescence. Bar graphs (right) are immunofluorescent quantification of active integrin β1 (left) or Talin1 (right) at cell surface of protrusions. All analyses of total fluorescence were normalized by cell area and at least 20 cells of each genotype were analyzed. Scale bars = 10 um. In all panels, statistics were one-way ANOVA with Tukey's post hoc test. Unless otherwise noted, **p<0.01, ***p<0.001, ns no significant difference.
DOI: https://doi.org/10.7554/eLife.45508.007

The following figure supplement is available for figure 3:

**Figure supplement 1.** DDR2 is required for full activation of collagen binding b1 Integrin in CAFs supplement 1.
DOI: https://doi.org/10.7554/eLife.45508.008

was less total RAP1 present (*Figure 4B*), and when corrected for the differing amounts of RAP1 protein there was significantly less activation of RAP1 in *Ddr2* depleted hCAFs after plating on collagen I (*Figure 4B*). To determine if defective collagen I-induced RAP1 activation by DDR2 was required and sufficient for Talin1 recruitment, we expressed two forms of dsRed-tagged constitutively activated RAP1 (caRAP1) in WT and *Ddr2* depleted hCAFs. Cells were plated on collagen I and dsRed positive cells identified (*Figure 4—figure supplement 1A*). When active β1 Integrin and Talin1 expression at cell surface of cell protrusions was determined by super resolution microscopy, both RAP1 <V12> and RAP1 <Q63E> rescued β1 Integrin surface activation as well as the recruitment of Talin1 to surface β1 Integrin in *Ddr2* depleted hCAFs (*Figure 4C*). In another approach WT and *Ddr2* depleted hCAFs were treated with Forskolin to activate cAMP which then activates RAP1, independent of cell surface receptors. As with caRAP1, Forskolin treatment rescued the defect in β1 Integrin activation at the cell surface of cell protrusions as well as Talin1 recruitment to the cell surface of cell protrusions in *Ddr2*-ve cells (*Figure 4—figure supplement 1B*).

These results indicated that following exposure to collagen I, DDR2 activation leads to upregulation of RAP1 activity which enhanced Talin1 activation and recruitment to integrin complexes in cellular protrusions and subsequent full activation of collagen-binding β1 integrins. These data also excluded a defect in the RAP1-Talin1-β1 Integrin signaling in *Ddr2* depleted hCAFs.

## Deletion of Ddr2 in breast tumor CAFs results in altered collagen fiber organization, decreased tumor stiffness, and is associated with decreased lung metastases

To determine if mechanotransduction regulation by DDR2 observed in breast tumor CAFs in ex vivo culture was relevant in an in vivo setting, we deleted the *Ddr2* gene in breast tumor CAFs and assessed the mechanical properties of primary tumors and the extent of lung metastasis. Selective genetic targeting of CAFs in vivo is difficult due to the now appreciated heterogeneity of tissue and tumor fibroblasts and the non-specificity of available cre drivers (*Costa et al., 2018*) (*Kalluri, 2016*). With these limitations in mind, we tested a number of cre expressing mice using an LSL-tdTomato reporter mouse. Our intent was to use the cre line that expressed in the majority of breast tumor CAFs (as defined by FAP staining). We tested Acta2-cre (αSMA-cre), Col1-cre, and S100a4-cre (FSP1cre) mouse lines. In MMTV-PyMT breast tumors we found that FSP1cre was expressed in the majority of CAFs (75% of FAP+ cells were red) (*Figure 5—figure supplement 1A*). Breast tumor epithelial-derived cells (luminal K8+ and basal K14+) and endothelial cells (CD31+) did not express FSP1cre (*Figure 5—figure supplement 1A*). As has been described previously, CD45+ leukocyte cells did express FSP1cre (90% of CD45+ cells were red) (*Figure 5—figure supplement 1A*).

Encouraged by these results, and cognizant of the lack of specificity of FSP1cre for solely CAFs, we generated FSP1cre; Ddr2^fl/fl; MMTV-PyMT mice (*Ddr2*^-/- FSP1cre mice) and appropriate controls. DNA analysis of the *Ddr2*^fl/fl allele in whole breast tumors from FSP1cre containing mice revealed

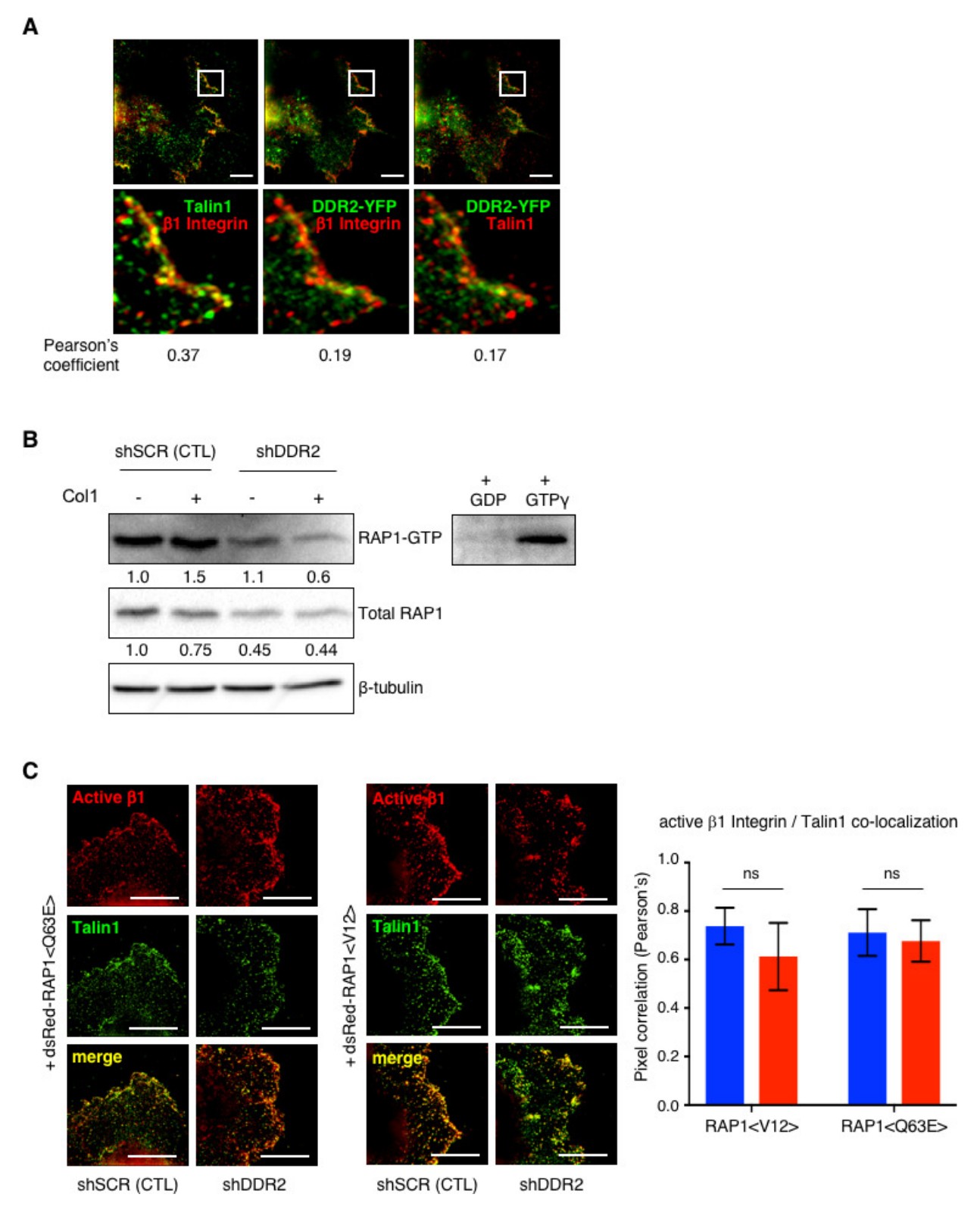

**Figure 4.** Collagen I stimulated DDR2 activates Rap1 that controls Talin1 activation and recruitment to collagen binding β1 Integrins. (**A**) n-SIM super resolution microscopy images of Talin1 (green) and total integrin β1 (red) (left panels), DDR2-YFP (green) and total integrin β1 (red) (middle panels), or DDR2-YFP (green) and Talin1 (red) (right panels) immunofluorescence in WT hCAFs plated on collagen I. All lower panels are a higher resolution image of white boxed region in upper panels. Co-localization was quantified by Pearson's coefficient which is listed below each set of panels. All analyses of

*Figure 4 continued on next page*

*Figure 4 continued*

total fluorescence were normalized by cell area and at least 20 cells were analyzed. Scale bars = 5 um. (**B**) RAP1 activation assay. RAP1-GTP pulldown with RAL GDS RAP1 binding domain attached to agarose was Western blotted with an anti-RAP1 antibody (upper panel), input control (5%) RAP1 level (middle panel) and loading control (lower panel). Control hCAF (shSCR) and hCAF depleted of *Ddr2* (shDDR2) cells were plated on collagen I (+) or no collagen (-) for 15 min. GDP and GTPγ negative and positive controls on right. Results were quantified by densitometry in ImageJ and normalized to β-tubulin and total RAP1. Shown is a representative experiment of 3 replicates. (**C**) Representative n-SIM super-resolution microscopy images of control hCAF (shSCR) and hCAF depleted of *Ddr2* (shDDR2) transfected with dsRed-RAP1 <Q63E> or dsRed-RAP1 <V12> plated on collagen I for 15 min, fixed, and stained for active integrin β1 (9EG7 - red) or Talin1 (green) antibodies. Scale bars = 10 um. Graph on right shows quantification of active integrin β1:Talin1 colocalization at cell surface of cell protrusions following introduction of constitutively active versions of RAP1. At least 20 cells were analyzed. In all panels, statistics were one-way ANOVA with Tukey's post hoc test. Unless otherwise stated, **p<0.01, ***p<0.001, ns - no significant difference.

DOI: https://doi.org/10.7554/eLife.45508.009

The following figure supplement is available for figure 4:

**Figure supplement 1.** Collagen I stimulated DDR2 activates Rap1 that controls Talin1 activation and recruitment to collagen binding b1 Integrins supplement 1.

DOI: https://doi.org/10.7554/eLife.45508.010

that the *Ddr2*[fl/fl] gene was rearranged but not in all cells, as expected (*Figure 5A*). Western blots of breast tumor CAFs isolated from these mice revealed that DDR2 protein was not expressed (*Figure 5B*). Deletion of *Ddr2* with FSP1cre significantly reduced the number of lung metastasis, without affecting primary tumor burden or latency (*Figure 5C–E*). The number of CAFs, as defined by PDFGRβ+ve cells, in *Ddr2*[-/-] FSP1cre primary breast tumors was not different from WT tumors (*Figure 5F*).

When 10 to 12 week old tumor slices were stained with picosirius red to assess collagen fiber content no significant difference in overall collagen content between WT and *Ddr2*[-/-] FSP1cre tumors at this stage of tumor development was noted (*Figure 5—figure supplement 1B*). However, when collagen fiber organization or architecture was analyzed by second-harmonic imaging (SHG) significant differences were noted. Compared to WT primary tumors, collagen fibers in *Ddr2*[-/-] FSP1cre primary tumors were curlier, shorter, and thinner (*Figure 5G*). This pattern corresponds to a benign or less aggressive tumor associated collagen signature (TACS1) of which there was a significant increase in *Ddr2*[-/-] FSP1cre tumors (*Figure 5H*). The more aggressive collagen signature associated with tumor cell invasion (TACS3) is the presence of thick collagen fibers perpendicular to the tumor-stromal interface, and these were more prevalent in WT primary tumors (*Provenzano et al., 2006*). In *Ddr2*[-/-] FSP1cre tumors there was a significant decrease in the TACS3 signature compared to wild type tumors (*Figure 5G and H*). When collagen organization within 5 microns of the tumor-stromal boundary was analyzed by focused ion beam scanning electron microscopy (FIB-SEM) there was less local collagen present, and the collagen fibers present were significantly thinner and more fragmented in *Ddr2*[-/-] FSP1cre tumors and ubiquitous *Ddr2*[-/-] tumors (*Figure 5I*; quantified in *Figure 5J*, also see *Video 1*, *Video 2*, and *Video 3*).

Since collagen fiber structure and organization within tissues and breast tumor stroma can impact mechanical properties we determined and contrasted the stiffness of WT and *Ddr2*[-/-] FSP1cre tumors using Atomic Force Microscopy (AFM). Overall, *Ddr2*[-/-] FSP1cre tumors were significantly less stiff (*Figure 6A and B*). Propidium Iodide staining of tumor slices was used to identify tumor nodules, tumor-stromal boundary, and stroma (*Figure 6C*). When the elastic modulus within tumor nodules were compared with that at the tumor-stromal boundary, stiffness changes were only noted at the tumor-stromal boundary (*Figure 6D*), a phenotype which has been linked to tumor aggression and invasion (*Acerbi et al., 2015*). Consistent with the change in overall tumor stromal stiffness, pMLC staining of tumor slices revealed significantly less overall pMLC activity in *Ddr2*[-/-] FSP1cre tumors (*Figure 5—figure supplement 1C*).

In summary, analyses of *Ddr2*[-/-] FSP1cre tumors, in which the *Ddr2* gene was deleted in the majority of CAFs and CD45+ leukocytes, revealed that fewer ECM collagen fibers were present at the tumor-stromal boundary and those present appeared thinner and fragmented. The functional consequence of these findings was tumors of diminished stiffness, with stiffness changes most prominent at the tumor cell-stromal boundary. These changes in the tumor stroma were associated with less lung metastases.

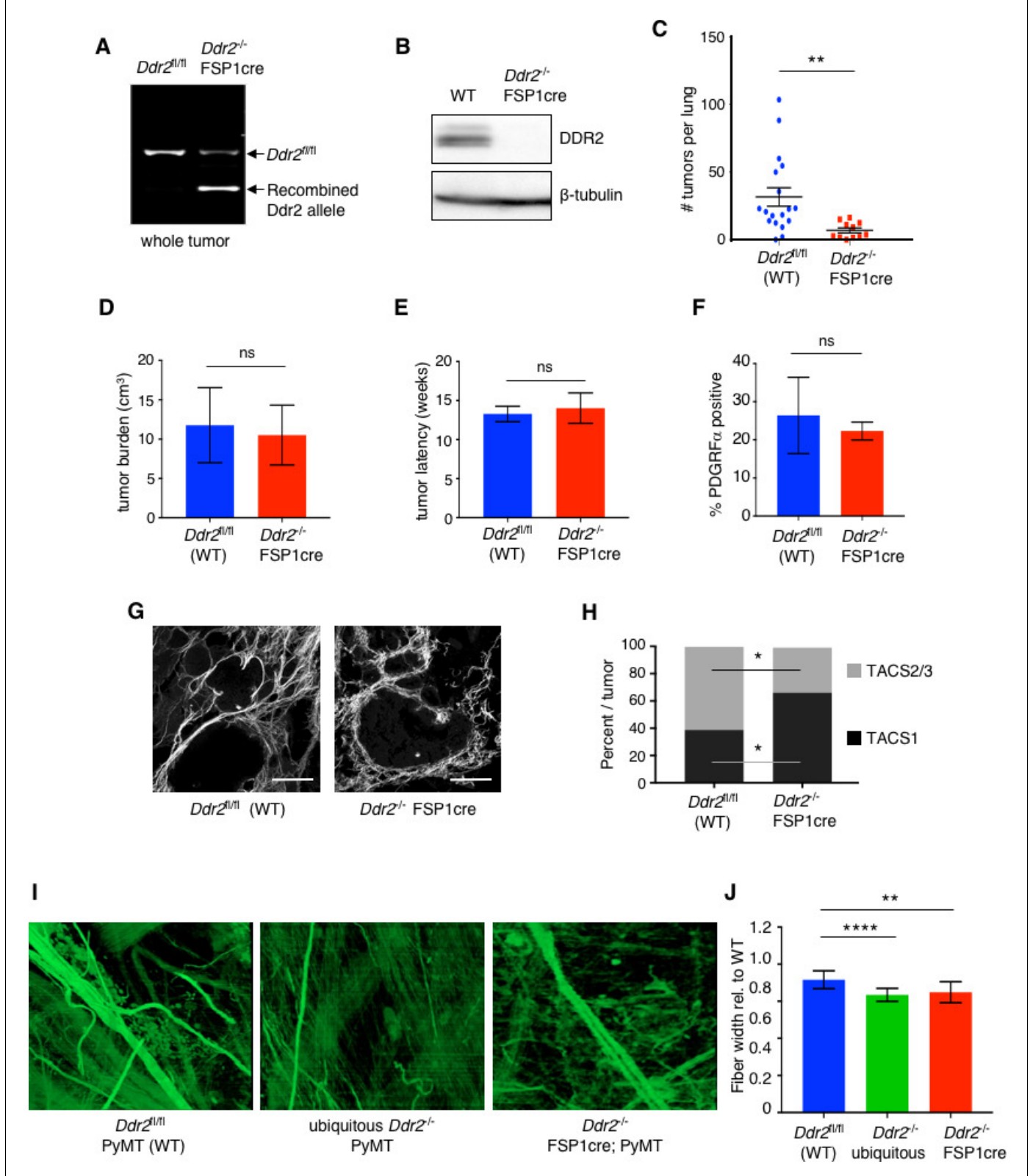

**Figure 5.** Deletion of *Ddr2* in breast tumor CAFs results in altered collagen fiber organization and decreased lung metastases. (**A**) DNA PCR genotyping of *Ddr2* alleles from whole tumors from indicated mice. (**B**) Western blot of CAFs isolated from MMTV-PyMT (WT) or FSP1cre; *Ddr2*^fl/fl; MMTV-PyMT (*Ddr2*^-/- FSP1cre) mice with the indicated antibodies. (**C**) Number of total lung metastases in MMTV-PyMT (WT) (n = 18 mice) or FSP1cre; *Ddr2*^fl/fl; MMTV-PyMT (*Ddr2*^-/-; FSP1cre) (n = 10 mice) mice at termination (@20 weeks or when a single tumor reached 2 cm). \*\*p=0.0095. All mice
*Figure 5 continued on next page*

Figure 5 continued

were >99% FVB/n. Lung tumors were enumerated histologically. (D) Primary tumor growth rate in MMTV-PyMT (WT) (n = 18 mice) or FSP1cre; $Ddr2^{fl/fl}$; MMTV-PyMT ($Ddr2^{-/-}$ FSP1cre) (n = 10 mice) as represented by time in weeks to end stage (single tumor 2 cm in largest dimension). (E) Total primary tumor burden in WT; MMTV-PyMT (WT) (n = 18 mice) or FSP1cre; $Ddr2^{fl/fl}$; MMTV-PyMT ($Ddr2^{-/-}$ FSP1cre) (n = 10 mice). Represented as the total volume of all primary tumors per mouse. (F) FACS quantification of percent PDGFRα positive cells (i.e., CAFs) per total primary tumor in size- and age-matched MMTV-PyMT (WT) or FSP1cre; $Ddr2^{fl/fl}$; MMTV-PyMT ($Ddr2^{-/-}$ FSP1cre) tumors. Four tumors each from four different mice were analyzed (total of 16 tumors). (G) Representative second harmonic images (SHG) of 10–12 week MMTV-PyMT (WT) or FSP1cre; $Ddr2^{fl/fl}$; MMTV-PyMT ($Ddr2^{-/-}$ FSP1cre) tumors. Scale bars = 100 um. (H) Quantification of TACS1 and TACS2/3 phenotype in tumor analyzed in (H). Six tumors from three different mice per genotype were analyzed (total of 18 tumors). *p<0.05. (I) 2D en face rendering of 3D reconstructed FIB-SEM images of tumor-stromal boundary (2–5 μM) from 12 week old MMTV-PyMT (WT), ubiquitous $Ddr2^{-/-}$ null; MMTV-PyMT, or $Ddr2^{-/-}$; FSP1cre; MMTV-PyMT tumors. Collagen fibers are green. Three tumors of each genotype were analyzed (n = 3 for each). See also *Videos 1*, *2* and *3*. (J) Quantification of fiber width in pixels relative to WT from FIB-SEM images in (I) for indicated tumors (n = 3 for each genotype). Fibers were quantified with CT-FIRE software (LOCI, Madison, WI). ****p<0.0001, **p=0.0041. In all panels, statistics were one-way ANOVA with Tukey's post hoc test. Unless otherwise noted, *p<0.05, **p<0.01, ***p<0.001, ns no significant difference.

DOI: https://doi.org/10.7554/eLife.45508.011

The following figure supplement is available for figure 5:

**Figure supplement 1.** Deletion of Ddr2 in breast tumor CAFs results in altered collagen fiber organization and decreased lung metastases supplement 1.

DOI: https://doi.org/10.7554/eLife.45508.012

## In vivo, the presence of DDR2 in primary breast tumor CAFs and tumor cells controls full β1 Integrin activation

To determine whether the action of DDR2 in breast tumor CAFs controlled β1 Integrin activity in vivo we freshly isolated primary MMTV-PyMT breast tumor organoids (200–500 cells) from WT, ubiquitous $Ddr2^{-/-}$, and FSP1cre; $Ddr2^{-/-}$ breast tumor bearing mice and immediately cultured these in 3D collagen I matrices under hypoxic conditions (*Hwang et al., 2019*). Tumor organoids were at no time exposed to plastic tissue culture plates. Tumor organoids were then stained with 9EG7 β1 Integrin antibody that recognizes activated Integrin. FAP staining was utilized to identify CAFs, and a control β1 Integrin antibody used to identify total β1 Integrin. In WT breast tumor organoids embedded within collagen I there was abundant active β1 Integrin present in both CAFs (FAP +ve) and tumor cells (FAP -ve) (*Figure 7A*; quantified in D and E). In contrast, in $Ddr2^{-/-}$ FSP1cre tumor organoids there was significantly decreased active β1 Integrin level in CAFs, while still present and unchanged in tumor cells that are $Ddr2^{+/+}$ in these mice (*Figure 7B*; quantified in D and E). In ubiquitous $Ddr2^{-/-}$ tumor organoids (all cells $Ddr2^{-/-}$) active β1 Integrin level was decreased in both CAFs and tumor cells (*Figure 7C*; quantified in D and E). These results were not a result of a decrease in β1 Integrin level in $Ddr2^{-/-}$ cells or a change in the number of CAFs present between differing genotypes (*Figure 7A–D*). These results indicated that the action of DDR2 in CAFs and tumor cells within primary breast tumors, in vivo, regulate β1 Integrin activation in response to collagen I exposure.

## Discussion

The action of the fibrillar collagen receptor DDR2 in CAFs, and possibly other cells (see later), within the primary stroma was found to be a critical regulator of breast tumor ECM collagen fiber organization and tumor stiffness. FSP1cre mediated deletion of *Ddr2* in MMTV-PyMT breast tumors resulted in tumors with reduced stiffness, particularly at the tumor-stromal boundary, and significantly altered collagen fiber organization again particularly at the tumor-stromal boundary. These changes were associated

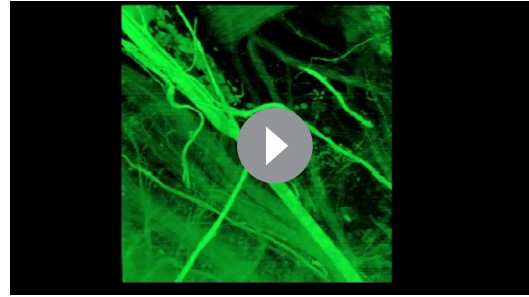

**Video 1.** FIB-SEM of WT PyMT tumor breast tumor cell-stromal boundary. This is a 360˚ depiction of a Focused Ion Beam Scanning Electron Micrograph (FIB-SEM) of the tumor-stromal boundary of a WT PyMT breast tumor. Collagen fibers are in green. Views are within 2 μm from the tumor cell surface.

DOI: https://doi.org/10.7554/eLife.45508.013

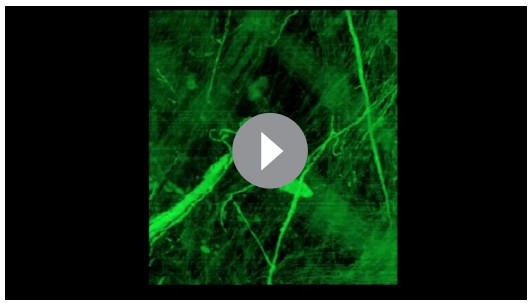

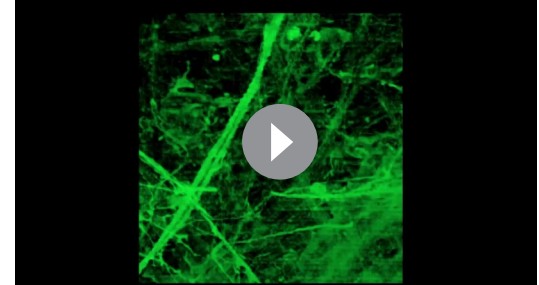

**Video 2.** FIB-SEM of ubiquitous *Ddr2*⁻/⁻ PyMT breast tumor cell-stromal boundary. This is a 360° depiction of a Focused Ion Beam Scanning Electron Micrograph (FIB-SEM) of the tumor-stromal boundary of a ubiquitous Ddr2⁻/⁻ PyMT breast tumor. Both tumor cells and stromal cells are deleted of *Ddr2*. Collagen fibers are in green. Views are within 2 μm from the tumor cell surface.

DOI: https://doi.org/10.7554/eLife.45508.014

**Video 3.** FIB-SEM of WT PyMT tumor. This is a 360° depiction of a Focused Ion Beam Scanning Electron Micrograph (FIB-SEM) of the tumor-stromal boundary of a Ddr2ᶠˡ/ᶠˡ; FSP1cre; PyMT breast tumor. Tumor stromal CAFs are deleted of *Ddr2*. Tumor cells are *Ddr2*⁺/⁺. Collagen fibers are in green. Views are within 2 μm from the tumor cell surface.

DOI: https://doi.org/10.7554/eLife.45508.015

with a significant decrease in lung metastases. Isolated CAFs from these mouse tumors lack expression of DDR2, as well as human breast tumor CAFs depleted of *Ddr2* exhibit decreased mechanotransduction properties. Their cell spreading on collagen I was decreased and formed smaller FAs without any change in the number of FAs, exhibited decreased collagen I gel contraction in 3D, generated reduced traction forces on collagen I coated hydrogels, and produced an altered collagen matrix. All these phenotypes could be reversed by re-expression of WT DDR2. DDR2 was found to be important for the full activation of collagen binding β1-containing Integrins in CAFs in culture and in CAFs and tumor cells within primary breast tumors in vivo. DDR2 influenced collagen binding β1 Integrin activity by regulating RAP1 mediated Talin1 and Kindlin2 activation and, or recruitment to Integrin complexes at surface adhesion sites. In vivo, breast tumors deleted of *Ddr2* in CAFs or ubiquitously in all cells within tumors exhibit decreased β1 Integrin activation in CAFs, and tumor cells (and CAFs), respectively.

In addition to regulating Integrin-based mechanotransduction, the action of DDR in CAFs also affects mRNA expression of collagen genes as well as collagen modifying enzymes such as lysyl oxidases and MMPs (*Corsa et al., 2016*) that can also contribute to altered matrix collagen fibers, tumor fibrosis and associated with enhanced metastasis (*Cox et al., 2013*). Indeed, FIB-SEM analysis (*Figure 5I and J*; *Video 1*, *Video 2*, and *Video 3*) revealed that within 2–5 microns of the tumor-stromal interface there was less collagen and the collagen fibers present in *Ddr2* deficient tumors were thinner and appeared more fragmented. Interestingly, corresponding trichrome or Sirius red histologic quantification of collagen fibers in the same tumors analyzed by FIB-SEM revealed no difference between *Ddr2*⁻/⁻ and WT tumors. SHG analysis of collagen fiber orientation, length, and thickness in the same tumors as above did note changes in collagen fiber structure when *Ddr2* was deleted from CAFs, like in the FIB-SEM result.

In breast tumors, FSP1cre is expressed not only in CAFs but in the majority of CD45+ myeloid cells. Thus, we cannot exclude a contribution of DDR2's action in myeloid cells as contributing to the altered CAF function in vivo or altered ECM remodeling in vivo. Inflammatory modulators secreted by cancer associated myeloid derived cells can affect tumor ECM directly (e.g., MMPs) or indirectly through cytokine secretion that activate CAFs (*Ruffell et al., 2012*). Moreover, DDR2 has been reported to be expressed in neutrophil cell lines where it regulates migration or chemotaxis through collagen fibers (*Afonso et al., 2013*), as well as in dendritic cell lines (*Poudel et al., 2012*). However, in ubiquitous *Ddr2*⁻/⁻ breast tumors there were no differences in the numbers of immune cell types observed compared to WT tumors (*Corsa et al., 2016*). The effect of reciprocal bone marrow transplantation of WT bone marrow into in *Ddr2*⁻/⁻; MMTV-PyMT mice or *Ddr2*⁻/⁻ bone marrow into WT MMTV-PyMT mice upon breast cancer metastasis and primary tumor ECM architecture and mechanical properties will address this possibility. Regardless, the accumulated cellular and in vivo data

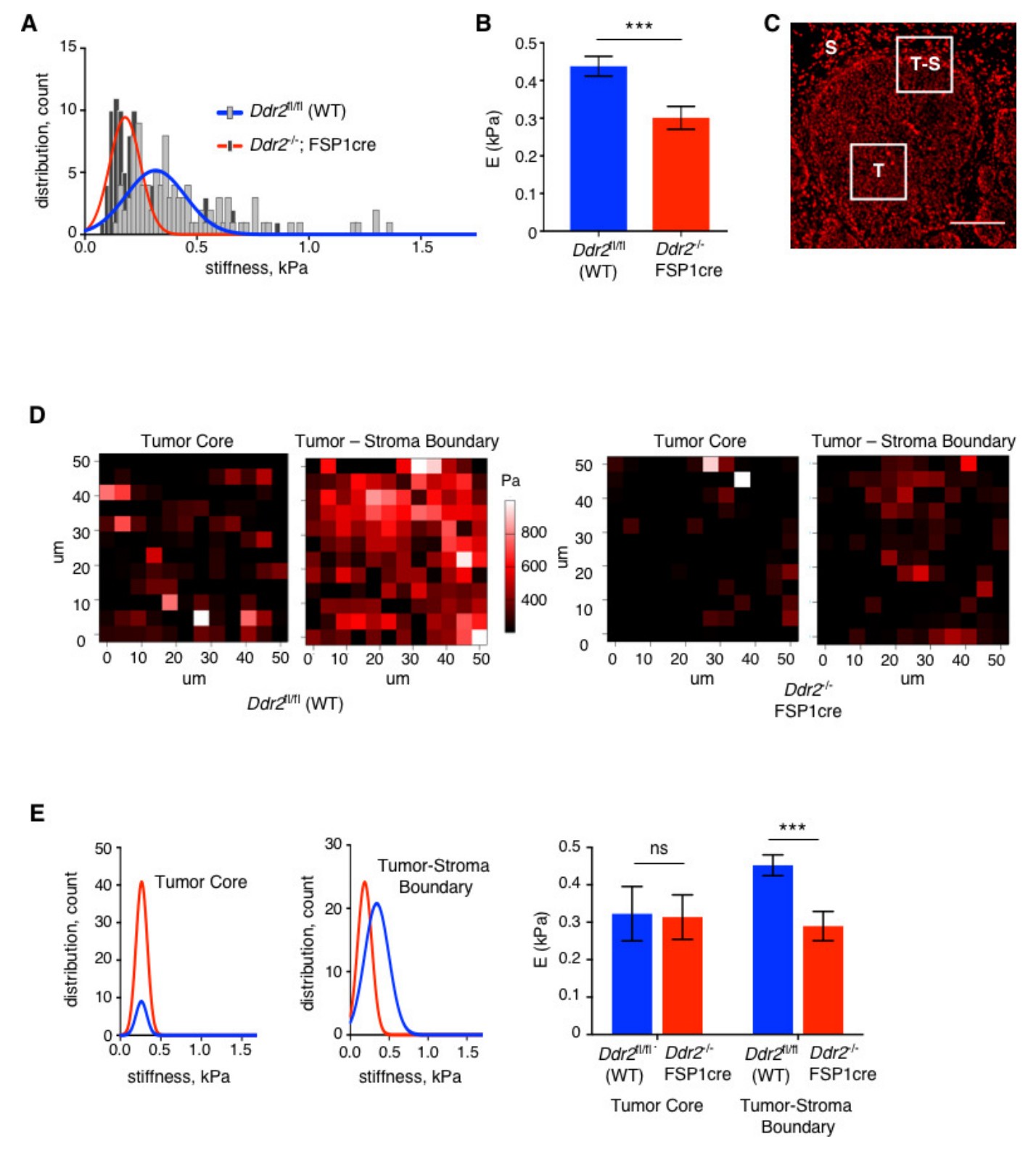

**Figure 6.** Deletion of *Ddr2* in breast tumor CAFs results in decreased tumor stiffness. (**A**) Compiled total tumor stiffness measurements represented as a histogram fit with a Gaussian curve (kPA, Young's elastic modulus) for WT PyMT (blue line) and *Ddr2*[-/-] FSP1cre (red line) tumors. Six (6) breast tumors from six different mice were analyzed for each genotype. (**B**) Bar graph quantifying average of tumor stiffness (elastic modulus) for WT PyMT primary breast tumors and *Ddr2*[-/-] FSP1cre primary breast tumors. Six (6) breast tumors from six different mice were analyzed as in (**A**) to generate the averages. *Figure 6 continued on next page*

*Figure 6 continued*

***p=0.0008. (C) Image of Propidium Iodine staining of a tumor slice that was used to identify tumor core (T), tumor-stromal boundary (T–S), and stromal compartments (S) regions used for measurements of stiffness. (D) Representative AFM stiffness heat map from MMTV-PyMT (WT) (left two panels) or FSP1cre; *Ddr2*fl/fl; MMTV-PyMT (*Ddr2*-/- FSP1cre) (right two panels) breast tumors. In each set of panels, tumor core (left) and tumor-stromal boundary (right) measurements are shown. (E) Compiled stiffness data of tumor core (left) or tumor-stroma boundary (middle) for MMTV-PyMT (WT – blue lines and blue columns) or FSP1cre; *Ddr2*fl/fl; MMTV-PyMT (*Ddr2*-/- FSP1cre - red lines and red columns) tumors. Data are represented as the Gaussian curve which was fitted to a histogram. Right is a bar graph showing averages of data for both tumor nodules and tumor-stromal boundary. ***p=0.0003. E = Young's elastic modulus. Six (6) tumors from six different mice per genotype were analyzed and data plotted are the top 100 best fitting curves per group. The Hertz method was used to calculate elasticity and Poisson's ratio of 0.5 was used to calculate Young's elastic modulus. In all panels, statistics were one-way ANOVA with Tukey's post hoc test. Unless otherwise noted, *p<0.05, **p<0.01, ***p<0.001, ns no significant difference.

DOI: https://doi.org/10.7554/eLife.45508.016

presented herein lend strong support for the mechanotransduction actions of DDR2 in CAFs as likely being a significant contributor to breast tumor mechanical properties (e.g., stiffness). We can exclude the action of DDR2 in tumor cells as contributing to breast tumor stiffness since deletion of *Ddr2* in breast epithelial cells only, with either MMTV-Cre or K14-Cre, did not affect primary tumor matrix production or organization (*Corsa et al., 2016*).

Growth factor RTK signaling can increase integrin expression, activation, and recycling. Integrin signals also affect growth factor RTK signaling (*Ivaska and Heino, 2011*). Most examples of growth factor RTK activation of Integrins occurs at the cell surface through complex formation with Integrins or possibly during endocytosis and recycling of receptor/Integrins (*Ivaska and Heino, 2011*). By super resolution confocal microscopy or co-immunoprecipitation experiments we did not find any supportive evidence that DDR2 and collagen binding Integrins form complexes in cells. Whether DDR2 is endocytosed and recycled, and if so, what the functional consequences are has not been determined. A report has suggested that DDR1b is endocytosed and recycled to the cell surface and that the onset of endocytosis precedes receptor phosphorylation, suggesting that signaling could occur in intracellular vesicles (*Mihai et al., 2006*). Overexpression of DDR2 in HEK293 cells, cells that do not endogenously express DDR2, increase α1β1 and α2βl integrin affinity without affecting the cell surface levels of either Integrin (*Xu et al., 2012*). The molecular basis for this observation and the functional significance were not determined.

During breast cancer development and progression endogenous DDR2 expression is upregulated in CAFs and appears to be critical for their activation (*Corsa et al., 2016*) (*Gonzalez et al., 2017*). Herein, we show that DDR2, in breast tumor CAFs, controls collagen binding Integrin activation by activating RAP1 to regulate Talin1 and Kindlin2 activity and, or recruitment to cell surface integrin. Talin1 and Kindlin2 activation and recruitment to cell surface β1 Integrin can be regulated by other positive and negative interactions (*Calderwood et al., 2013*) (*Das et al., 2014*), but since constitutively activated RAP1 and Forskolin treatment rescued the defect almost entirely it is less likely that DDR2 influences these pathways. Precisely how DDR2 activates RAP1 remains to be determined. Inside-out RAP1 signaling regulating Talin1 mediated integrin activation has been well described and shown to be important in hematopoietic cells (*Lagarrigue et al., 2016*), and cultured cells (*Calderwood et al., 1999*). But in mesenchymal cells within tissues, in vivo, where cells such as CAFs are embedded in the presence of excess Integrin ligand (e.g., collagen I), it has been more difficult to ascertain whether the contribution of inside-out signaling to integrin activation is present and important (*Klapholz and Brown, 2017*). Herein we show that in the absence of full Talin1 activation in response to collagen activated DDR2 there are clear functional sequelae in mesenchymal cells in culture and in tumor progression in vivo.

Many, if not most, of the observed defective mechanobiologic properties of *Ddr2*-/- CAFs in various culture systems and *Ddr2*-/- tumors in vivo are likely to be manifestations of defective collagen binding β1 integrin activity rather than other DDR2 signaling pathways. DDR2 signaling also regulates the protein level, subcellular localization, and action of SNAIL1, an important mesenchymal cell transcriptional regulator (*Zhang et al., 2013*) (*Stanisavljevic et al., 2015*). SNAIL1 is a critical regulator of the fibrogenic response of fibroblasts and CAFs (*Stanisavljevic et al., 2015*) (*Zhang et al., 2016*). Moreover, mechanical signals (e.g., matrix stiffness) also regulates SNAIL1 levels and function (*Zhang et al., 2016*). As a result, the overall tumor matrix organization controlled by DDR2 signaling

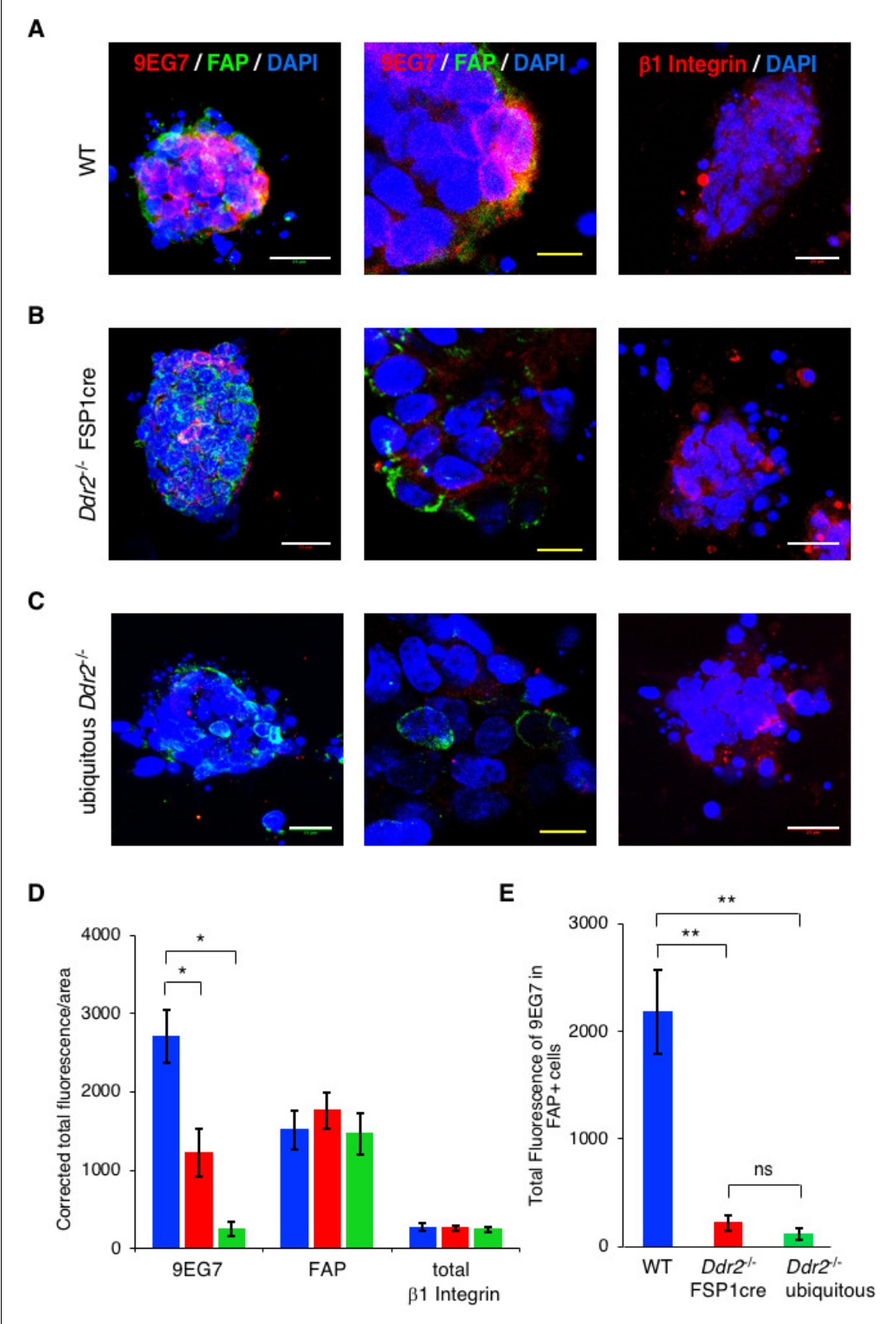

**Figure 7.** In vivo, the presence of DDR2 in primary breast tumor CAFs controls full β1 Integrin activation. WT (**A**), $Ddr2^{-/-}$ FSP1cre (**B**), and ubiquitous $Ddr2^{-/-}$(**C**) breast tumor organoids were isolated from 12 week primary tumors and immediately placed in a 3D collagen I microfluidic device under hypoxic conditions. After 48 hr immunofluorescence staining for active β1Integrin (red) and FAP (CAFs -green) (left and middle panels) or total β1Integrin (red) (right panel) were performed on organoids in the device. Organoids were also stained with DAPI to identify nuclei. The middle panels

*Figure 7 continued on next page*

*Figure 7 continued*

are higher resolution images of parts of the left panels. (**D**) Quantification of total fluorescence per tumor organoid for indicated antibodies in each set. Blue columns – WT tumor organoids; red columns - *Ddr2*^-/- FSP1cre tumor organoids, green columns - ubiquitous *Ddr2*^-/- tumor organoids. (**E**) Quantification of total fluorescence of active β1 integrin (9EG7) in only FAP+ cells (i.e., CAFs) in WT (blue), *Ddr2*^-/- FSP1cre (red), and Ubiquitous *Ddr2*^-/- tumor organoids. From 2 different mice 10–20 tumor organoids per genotype were analyzed. In all panels, statistics were one-way ANOVA with Tukey's post hoc test. Unless otherwise noted, $*p<0.05$, $**p<0.01$, $***p<0.001$, ns - no significant difference.

DOI: https://doi.org/10.7554/eLife.45508.017

in tumors, and CAFs specifically, likely involves both its regulation of collagen binding β1 Integrin and SNAIL1. Activation of DDR2 by fibrillar collagens, as defined by tyrosine phosphorylation of the cytoplasmic tail, is slow (hours) and sustained (hours - days) for reasons that are unclear (*Fu et al., 2013*). DDR2 signals that stabilize SNAIL1 protein in CAFs takes 4–6 hr of exposure to collagen I in cultured cells (*Zhang et al., 2016*). In contrast regulation of collagen binding β1 integrin by DDR2 was apparent within minutes (15–30 min) yet requires the kinase activity of DDR2. This suggests that the DDR2 signaling pathway regulating RAP1 and integrin activity could be distinct from that regulating SNAIL1 activation.

In summary, we provide evidence and mechanism for the action of the collagen binding receptor DDR2, primarily in cancer associated fibroblasts, as a critical pathway controlling the tumor permissive ECM that forms in aggressive breast cancers and that facilitates or supports tumor cell invasion, migration and metastasis. We also show that inside-out regulation of collagen binding Integrin occurs in mesenchymal cells and in vivo and can have significant impact upon cancer progression to metastasis. As such DDR2 could represent an important therapeutic target to prevent cancer metastasis. Our results also shed light into how the action of DDR2 in tissue fibroblasts may contribute to other organ fibrosis pathologies in response to injury or inflammation (*Zhao et al., 2016*) (*DeLeon-Pennell, 2016*).

## Materials and methods

### Cells utilized

Immortalized human breast tumor cancer associated fibroblasts (hCAFs) were kindly provided by Dr. S. McCallister (Harvard Medical School). Generation of control shSCR human breast tumor CAFs and hCAF cells depleted of *Ddr2* with shRNA (shDDR2) have been described previously (*Zhang et al., 2016*). Primary MMTV-PYMT tumor cells were isolated as previously described (*Corsa et al., 2016*). To isolate primary mouse breast tumor cancer associated fibroblasts (mCAFs), MMTV-PyMT breast tumors were dissected, minced, and minced pieces transferred to ~20 mL of digestion media per tumor (DMEM, 1% fbs, 0.2% Collagenase A (Roche), 0.2% trypsin (Gibco 27250–018), 50 µg/mL gentamycin, 5 µg/mL insulin) and rocked at 37 degrees for 30–45 min. The digested tissue was then washed twice with serum free media and treated with DNAse for 5 min at room temperature. Tissue was resuspended in ice-cold serum free media and serially centrifuged four times. Single cell fractions were collected and plated for 25–30 min in DMEM, 10% fbs at 37 degrees Celsius, 5% $CO_2$, 20% $O_2$. CAFs will adhere to the plate while other cells will not. The supernatant and non-adherent cells were removed and CAFs were maintained in DMEM, 10% fbs at 37 degrees Celsius, 5% $CO_2$, 20% $O_2$. All spontaneously immortalized primary CAF cell lines were submitted to FACS with PDGFRα antibodies. All cell lines utilized were Mycoplasma free as determined by Q-PCR analyses every 6 months.

### ECM synthesis by cultured CAFs and analysis

Human CAFs were plated to confluence on 12 mm glass coverslips in DMEM supplemented with 10% FBS and 50 µg/ml ascorbic acid, and media was changed daily for 7 days. Cells were then lysed on day 7 (25 mmol/L Tris-HCl, pH 7.4; 150 mmol/L sodium chloride; 0.5% Triton X-100; and 20 mmol/L ammonia hydroxide) for 3–5 min. Cellular debris was carefully washed away with 1X PBS. Resultant cell free ECMs were fixed in 4% paraformaldehyde for 15 min at room temperature and then blocked with 5% FBS in 1X PBS. ECMs were then incubated in mouse anti-fibronectin antibody (diluted 1:100, BD Biosciences) overnight at four degrees, washed twice, and then incubated in goat

anti-mouse AlexaFluor 488 secondary (diluted 1:500, Life Technologies), washed four times, mounted in Vectashield (VWR, 101098–044), and sealed with nail polish. Immunofluorescence was analyzed on a confocal microscope (LSM 700; Carl Zeiss, Jena, Germany) at room temperature with Zen 2009 software. ImageJ was used to adjust brightness and contrast.

## Collagen gel contraction assay

two $\times$ $10^5$ CAFs were embedded in 100 ul of 1 mg/ml collagen one gel (Rat tail collagen, Corning CB354249) which was then spread with a pipet tip into the well of glass bottom 12 mm Mattek dishes. The gel was allowed to solidify at 37 degrees Celsius for 20 min after which 2 mL of DMEM+ 10% fbs was added and gels were gently detached with a pipet tip. Gels were imaged after 3 days, and percent contraction was calculated relative to initial gel area by tracing in ImageJ.

## Western blotting

Cells were lysed in 1X RIPA buffer supplemented with 1 mM PMSF, 1 mM sodium vanadate, 1 mM sodium fluoride, and 10 ug/ml each aprotinin and leupeptin. Lysates were sonicated twice for 30 s and centrifuged at 14,000 RPM, 10 min. Cleared lysates were separated by SDS-PAGE, transferred onto PVDF membrane, and blocked for 1 hr at room temperature in 5% non-fat dry milk, 1X TBS-0.5% Tween. Membranes were incubated in primary antibody overnight at four degrees with gentle agitation, washed twice with TBS-0.5% Tween, and incubated with anti-mouse or anti-rabbit HRP secondary antibody for one hour at room temperature. Membranes were then washed four times with TBS-0.5% Tween and developed with ECL (Pierce, 32106). For use of specific antibodies see *Table 1*.

## Co-Immunoprecipitation

Tissue culture dishes were coated with 50 µg/mL collagen in water and allowed to dry overnight at room temperature. The next day, dishes were blocked with 1% BSA for 1 hr at room temperature and then sterilized under UV light for 20 min. Cells were serum starved overnight and then removed from plates non-enzymatically. Cells were allowed to adhere for 1 hr and then lysed in 20 mM Tris, pH 7.5, 1% Triton X-100, 0.1% SDS, 150 mM $CaCl_2$ supplemented with 1 mM PMSF, 1 mM sodium

**Table 1.** Antibodies used for Western blot, IF, and IP.

| Antibody | Source | Application | Concentration |
| --- | --- | --- | --- |
| A-SMA (1A4) | Sigma A2547 | WB | 1:10,000 |
| β-actin | Sigma A5441 | WB | 1:10,000 |
| β-tubulin | Sigma T4026 | WB | 1:2000 |
| CD31 | Abcam ab28364 | IF | 1:100 |
| CD45 | BD Biosciences 550539 | IF | 1:100 |
| Collagen 1a1 | EMD Millipore AB765P | IF | 1:200 |
| DDR2 | CST 12133 | WB | 1:1000 |
| FAP | EMD Millipore ABT11 | IF | 1:100 |
| Integrin β1 | BD Biosciences 552828 | IF | 1:50 |
| Integrin β1 | CST 4706 | WB | 1:500 |
| Integrin β1 (9EG7) | BD Biosciences 553715 | IF | 1:50 |
| K14 | Covance PRB-155P | IF | 1:100 |
| K8 | DSHB TROMA-1 | IF | 1:100 |
| p-MLC (Ser19) | CST 3671 | IF | 1:50 |
| PDGFRα-FITC | eBioscience 11-1401-80 | FACS | 1:50 |
| Rap1 | EMD Millipore 07–916 | WB | 1:500 |
| Talin1 (8D4) | Sigma T3287 | IF, WB | 1:250, 1:2000 |
| Vinculin | Sigma V9131 | IF | 1:250 |

DOI: https://doi.org/10.7554/eLife.45508.019

vanadate, 1 mM sodium fluoride, and 10 μg/ml each aprotinin and leupeptin. Equal amounts of protein were pre-cleared with Protein G Sepharose beads and then incubated with talin1 antibody overnight with gentle agitation. Protein G Sepharose beads were added for 1 hr at four degrees, then beads were washed four times with Co-IP buffer, resuspended in 2X Laemmli sample buffer, boiled, and separated by SDS-PAGE.

## RAP1 activation assay

For RAP1-GTP immunoprecipitation, we followed the manufacturer's instructions (EMD Millipore 17–321). Briefly, cells were serum starved overnight and then plated on collagen coated tissue culture dishes for 15 min. They were lysed in the supplied lysis buffer supplement with 1 mM PMSF, 1 mM sodium vanadate, 1 mM sodium fluoride, and 10 ug/ml each aprotinin and leupeptin. Samples were sheared with passages through 27G needle, spun, and supernatants incubated with RAL-GDS-RBD agarose beads at 4 degrees for 45 min. Beads were washed and resuspended in 2X Laemmli buffer, boiled, separated by SDS-PAGE, and then Western blotted with anti-RAP1 antibodies.

## Traction force measurement

Glass coverslips were activated with 3-APTMS for 5 min and fixed in 0.5% glutaraldehyde for 30 min at room temperature. Hydrophobic coverslips were made by treatment with Sigmacote. Soft (792 Pa) polyacrylamide hydrogels were made by polymerizing (final concentrations of 5% acrylamide and 0.1% bis-acrylamide with 0.5% dark red fluorescent beads, 0.2 um; Thermo Fisher Scientific F8807) gel in a sandwich between the functionalized and hydrophobic coverslips. Gels were allowed to polymerize for 30 min at room temperature, the sandwich separated and washed. The surface of the gel was functionalized with 0.5 mg/mL sulfo-SANPAH in 50 mM HEPES, pH 8.2 under UV light for 10 min. Gels were extensively washed and then incubated with 50 ug/mL collagen 1 in 50 mM HEPES, pH 8.2 overnight at four degrees Celsius. The next day, gels were washed and equilibrated in DMEM. Cells were plated sparsely and allowed to adhere and spread overnight. During microscopy, cells were kept at 37 degrees and under 5% $CO_2$ in an incubated plate holder. Images were taken before and after trypsinization, and bead displacements calculated with a Matlab program.

## Super resolution microscopy

For collagen coating, 50 ug/mL collagen in water was spread on 12 mm glass coverslips (no. 1.5, high precision) and allowed to dry at room temperature overnight. The next day, the coverslips were blocked with 1% BSA in PBS for 1 hr at room temperature and sterilized under UV for 20 min. Cells were serum starved and removed from tissue culture plates non-enzymatically. They were then sparsely plated ($1 \times 10^4$ per coverslip) and allowed to adhere and spread for indicated times. Cells were then fixed in 4% paraformaldehyde for 15 min at room temperature. Cells were permeabilized in 0.1% Triton X-100 in PBS for 5 min at room temperature, washed with PBS, and blocked with 5% normal goat serum in PBS. Primary antibodies were added and incubated at four degrees Celsius overnight. Coverslips were washed and secondary fluorescent antibody added for 1 hr at room temperature. Coverslips were washed again and mounted in Prolong Diamond mounting medium. Coverslips were allowed to cure for 24 hr. For focal adhesion quantification, cells were imaged by confocal microscopy on NIS-Elements software (Nikon A1Rsi, inverted). Z-stacks were taken with a step size of 0.2 um with a 40X objective. Z-stacks were flattened by maximum intensity projection, and focal adhesions were quantified in ImageJ by subtracting the background, thresholding to the same level for all samples, and running particle analysis. For n-SIM super-resolution microscopy, images were taken with NIS-Elements software on a Nikon Ti-E microscope with a high NA 100X objective. Fluorescence was captured with an Andor Zyla 4.2 Megapixel sCMOS camera. Z-stacks were taken for all images with a step size of 0.15 um. n-SIM images were reconstructed in NIS-Elements, and fluorescence intensity co-localization quantification was done in ImageJ. For use of specific antibodies see *Table 1*.

## Mouse tumor/metastases assays

### Spontaneous genetic model

The conditional *Ddr2* floxed allele was generated as previously described (*Corsa et al., 2016*). This allele was crossed to FSP1cre; MMTV-PyMT mice to generate FSP1cre; *Ddr2*fl/fl; MMTV-PyMT mice.

Wild type littermates were used as controls. All genotyping was done on DNA from mouse tails. DNA was extracted and PCR run using KAPA Biosystems HotStart PCR (KK5621). For primers used see *Table 2*. FSP1cre mice, on an FVB/n background, were from the Dr. Zena Werb (San Francisco, CA). Tumor bearing mice were monitored weekly until end stage (a single tumor of 2 cm), typically 18–20 weeks. Mice were euthanized and primary tumors and lungs collected. All mice analyzed were >90% FVB/n. Lungs were fixed overnight in 10% neutral buffered formalin and then embedded in paraffin. Three 5 um sections were taken 200 µm apart per lung and stained with hematoxylin and eosin. Metastases were counted in all lobes and documented as average number of total lung metastases.

### Orthotopic transplant model

8 week old female $Ddr2^{+/+}$ or ubiquitous $Ddr2^{-/-}$ (on a Balb/C background >6 generation) received breast transplants (mammary fat pad) of $10^6$ 4T1 breast tumor cells (Balb/C and express DDR2). After 2 weeks primary tumor volume was determined at autopsy and number of lung tumors per slice from all five lobes identified and counted histologically.

### Lung colonization assay

$10^6$ primary PyMT breast tumor cells or $10^5$ 4T1 breast tumor cells were injected I.V. (tail vein) into respective syngeneic mice. After 2 or 1 week, respectively, the number of lungs tumors were identified and enumerated histologically.

## Tumor immunofluorescence

Tumors were dissected away from the skin and then cut into <1 cm pieces to allow efficient fixation. Tumors were fixed overnight in 10% neutral buffered formalin and then equilibrated in 30% sucrose overnight at four degrees. Equilibrated tissues were embedded in OCT and cryosectioned at 5–10 um per section. Sections were post-fixed in 4% paraformaldehyde for 15 min, permeabilized in 0.1% Triton X-100 for 5 min. and blocked in 5% goat serum for 1 hr at room temperature with washes in 1X PBS in between each step. Primary antibodies were incubated overnight at four degrees. Sections were then washed twice with 1X PBS and secondary antibody added for 1 hr at room temperature. Sections were then washed four times in 1X PBS, mounted in VectaShield with DAPI (VWR, 101098–044), and sealed with nail polish. Images were taken on an inverted Nikon epifluorescence microscope. Brightness and contrast adjustment as well as co-staining quantification was done manually in ImageJ. For use of specific antibodies see *Table 2*.

## Mouse tumor organoid isolation, culture, and analyses

Tumor bearing mice were monitored weekly and euthanized at 12 weeks. All mice were used in compliance with the Washington University Institutional Animal Care and Use Committee under protocol #20150145. Mice mammary tumor organoids were obtained as previously described (*Corsa et al.,*

**Table 2.** Primer sequences for DDR2 mouse genotyping.

| Name | Sequence (5' – 3') |
| --- | --- |
| FRT 5' Fwd | CTGTGTCTCTGGCTCAAAGTGTC |
| Targeted exon Rv | CCTTCCCAAGGCAGACCATTC |
| PyMT Fwd | GGAAGCAAGTACTTCACAAGGG |
| PyMT Rv | GGAAAGTCACTAGGAGCAGGG |
| Cre Fwd | GCATTACCGGTCGATGCAACGAGTGATGAG |
| Cre Rv | GAGTGAACGAACCTGGTCGAAATCAGTGCG |
| ROSA-LSL-TdTomato Fwd | GAGGGCCGCCACCACCTGTTCCTGTACGG |
| ROSA-LSL-TdTomato Rv | ATGATACAAAGGCATTAAAGCAGCGTATCC |
| ROSA-WT Fwd | GGGGAGTGTTGCAATACCTTTCTGGGAGTTC |
| ROSA-WT Rv | AAAACCGAAAATCTGTGGGAAGTCTTGTC |

DOI: https://doi.org/10.7554/eLife.45508.018

*2016*), mixed with 2 mg/ml collagen I solution, loaded into the middle tissue chamber of a microfluidic device, allowed to polymerize (37°C, 20% $O_2$), and media (DMEM, 10% FBS, P/S) was delivered to the top and bottom fluidic lines and cultured in 5% $O_2$ for 48 hr. All immunostaining was performed with organoids maintained within the devices, and all reagents were delivered via microfluidic lines. After fixing and blocking, organoids were stained for 9EG7, total Integrin β1, and FAP; all primary antibody staining was incubated overnight at 4C. Species-specific secondary antibodies (488 or 566 wavelength) and nuclei staining (DAPI) were also used. Imaging was performed via confocal microscopy (Zeiss, 63X). Analysis was performed using FIJI to quantify fluorescence (corrected total cell fluorescence (CTCF) = Integrated Density – (Area of selected cell X Mean fluorescence of background readings)) in the entire tumor organoid as well as in FAP positive cells only.

## Second harmonic generation and TACS scoring

10 to 12 week old tumors were dissected and fixed in 10% neutral buffered formalin overnight at room temperature. They were then embedded in paraffin and sectioned in 5–10 um sections. In some cases, sections were stained with H and E, picosirius red, or trichrome stain prior to SHG imaging. Prior staining had no effect on SHG signal. Images were acquired on a Zeiss LSM 880 Airyscan confocal microscope using an inverted, motorized Zeiss Axio Observer Z1 frame. Two-photon images were collected at 880 nm, using non-descanned detectors set to 440 nm for SHG. Three to four z-stacks were acquired (step size 2 um) per tumor. The z-stacks were compressed and TACS signature was scored by three blinded reviewers as previously described (*Corsa et al., 2016*) (*Provenzano et al., 2006*).

## Focused ion beam scanning electron microscopy (FIB-SEM)

Mice were perfused with pre-warmed, 37-degree, Ringer's solution (155 mM NaCl, 3 mM KCl, 2 mM $CaCl_2$, 1 mM $MgCl_2$, 3 mM $NaH_2PO_4$, 5 mM HEPES, pH 7.4, 10 mM glucose) for 2 min and then for 5 min with pre-warmed, 37-degree fixative (2.5% glutaraldehyde, 2% paraformaldehyde, 0.05% ruthenium red, 0.2% tannic acid in 0.15M cacodylate). Tumors were then dissected out and placed in fixative for 15 min at 37 degrees, then four degrees overnight. Samples were embedded in resin and scanned by FIB-SEM.

## Atomic Force Microscopy

Non-necrotic 10 to 12 week old tumors were gently dissected away from the skin and flash frozen in OCT. Tumors were sectioned at 20 μm per section. Just prior to AFM, tissues were quickly thawed in 1XPBS at room temperature and then maintained in 1X PBS supplemented with protease inhibitor cocktail (Roche Diagnostics, 11836170001) and propidium iodide (20 μg/ml). 5–6 force maps were taken of at least two tumors from three mice per group. AFM was performed as described (*Acerbi et al., 2015*). All indentations were taken on an MFP-3D-BIO AFM (Asylum Research) mounted on an Olympus X711 inverted fluorescent microscope in an TMC acoustic noise enclosure. We used silicon nitride cantilever tips with a 5 μm borosilicate glass sphere affixed to the tip with a spring constant of 0.06 N/m (Novascan, Boone, IA). The cantilever was calibrated with thermal oscillation prior to each experiment. Indentations were taken at 20 um/second loading-rate with a maximum force of 5 nN, and force maps were generated using the FMAP function on IGOR software (Asylum Research). The Hertz method was used to calculate elasticity and Poisson's ratio of 0.5 was used to calculate Young's elastic modulus.

## Statistical analysis

P-values were calculated using either Student's unpaired, two-tailed T-Tests or ANOVA with Tukey's post hoc, as noted in figure legends.

## Acknowledgements

This work was supported by NIH grants R01 CA196205, R01 CA223758, and U54 CA210173 (GDL), F30 CA200386 (SVHB), T32 GM07200 (WRG), and T32 CA113275 (CEB). PYH was supported by an American Cancer Society Postdoctoral Fellowship, and a WM Keck Foundation Postdoctoral Fellowship. We also gratefully acknowledge assistance in imaging provided the Washington University

Center for Cellular Imaging (WUCCI), which is supported by Washington University School of Medicine, The Children's Discovery Institute of Washington University and St Louis Children's Hospital (CDI-CORE-2015–505) and the Foundation for Barnes-Jewish Hospital (3770).

## Additional information

### Funding

| Funder | Grant reference number | Author |
| --- | --- | --- |
| National Institute for Health Research | R01 CA196205 | Gregory D Longmore |
| National Institute for Health Research | R01 CA223758 | Gregory D Longmore |
| National Institute for Health Research | U54 CA210173 | Gregory D Longmore |
| American Cancer Society | 131342-PF-17-238-01-CSM | Priscilla Y Hwang |
| National Institute for Health Research | F30 CA200386 | Samantha VH Bayer |
| National Institute for Health Research | T32 GM07200 | Samantha VH Bayer Whitney R Grither |
| National Institute for Health Research | T32 CA113275 | Craig E Barcus |

The funders had no role in study design, data collection and interpretation, or the decision to submit the work for publication.

### Author contributions

Samantha VH Bayer, Conceptualization, Data curation, Formal analysis, Investigation, Methodology, Writing—original draft, Conceived the project, designed the experiments, and wrote the manuscript; Whitney R Grither, Audrey Brenot, Priscilla Y Hwang, Conceptualization, Data curation, Formal analysis, Investigation, Methodology, Performed experiments and data analysis; Craig E Barcus, Data curation, Formal analysis, Investigation, Methodology, Generated reagents critical for experiments; Melanie Ernst, Patrick Pence, Data curation, Formal analysis, Investigation, Methodology, Performed experiments and data analysis; Christopher Walter, Data curation, Formal analysis, Assisted with acquisition and analysis of AFM data; Amit Pathak, Data curation, Formal analysis, Supervision, Funding acquisition, Project administration, Assisted with acquisition and analysis of AFM data; Gregory D Longmore, Conceptualization, Supervision, Funding acquisition, Project administration, Writing—review and editing, Conceived the project, designed the experiments, and wrote the manuscript, Supervised the research

### Author ORCIDs

Melanie Ernst (iD) https://orcid.org/0000-0002-8995-3507
Gregory D Longmore (iD) https://orcid.org/0000-0001-7568-8151

### Ethics

Animal experimentation: This study was performed in strict accordance with the recommendations in the Guide for the Care and Use of Laboratory Animals of the National Institutes of Health under protocol #20150145.

### Decision letter and Author response

Decision letter https://doi.org/10.7554/eLife.45508.022
Author response https://doi.org/10.7554/eLife.45508.023

## Additional files

### Supplementary files
- Transparent reporting form

DOI: https://doi.org/10.7554/eLife.45508.020

### Data availability

All data generated or analysed during this study are included in the manuscript and supporting files.

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
