## [Decision Letter]

Thank you for submitting your article "DDR2 controls breast tumor stiffness and metastasis by regulating Integrin mediated mechanotransduction in CAFs" for consideration by *eLife*. Your article has been reviewed by two peer reviewers and the evaluation has been overseen by a Reviewing Editor and Jonathan Cooper as the Senior Editor. The reviewers have opted to remain anonymous.

The reviewers have discussed the reviews with one another and the Reviewing Editor has drafted this decision to help you prepare a revised submission. While the work performed demonstrating the changes in biology in the primary tumor is of sufficient interest, there are some significant issues with the claims about metastasis and with image quantification, as outlined below:

1) The metastasis part is weak, and the emphasis and claims on metastasis are unwarranted. The claims need to be toned down to state that CAF DDR2 expression is important in tumor progression. Although the paper includes analysis of tumor growth in the lungs, this is based only on tail vein injection assays, which are called "metastasis" but are not. Moreover, there is very little mechanistic work connecting the biology of the primary tumor with development of lung metastases.

2) In the text, the authors contrast a published role for DDR2 in breast cancer lung metastasis in the presence of a primary tumor only (reciprocal orthotopic syngeneic breast tumor transplant experiments, Corsa CA 2016) with reciprocal tail vein injection in *Ddr2^+/+^* vs. *Ddr2^-/-^* host mice in which metastasis was unaffected by DDR2. Although these orthotopic data are previously published, they are foundational to the authors' overall claim regarding lung metastasis. The authors should repeat the orthotopic experiment and present this alongside the data in Figure 1.

3) Figure 3—figure supplement 1. Does Kindlin-2 remain at the focal adhesion sites in the shDDR2 cells? A co-localization of Kindlin-2 with total β1 integrin would help address this.

4) The images in Figure 7 from the tumor organoids are difficult to interpret. It seems that quantification was carried out in 3D z-stacks. The authors state that the active β1 integrin signal was reduced in CAFs but not in tumor cells in the *Ddr2^-/-^* organoids. It is difficult to see how this assumption can be made based on the type of images shown. Furthermore, the quantification represents total signal throughout the organoid.

---

## [Author Response]

1) The metastasis part is weak, and the emphasis and claims on metastasis are unwarranted. The claims need to be toned down to state that CAF DDR2 expression is important in tumor progression. Although the paper includes analysis of tumor growth in the lungs, this is based only on tail vein injection assays, which are called "metastasis" but are not. Moreover, there is very little mechanistic work connecting the biology of the primary tumor with development of lung metastases.

In original Figure 5(A-F) and Figure 5—figure supplement 1A we presented results from an experiment using a spontaneous model of breast cancer metastasis with high metastatic penetrance (MMTV-PyMT) in which deletion of Ddr2^fl/fl^ with FSP1-Cre (CAF deletion) resulted in a significant decrease in lung metastases (Figure 5C). In this experimental setup *Ddr2* is deleted in the majority of FAP+ CAFs and myeloid cells (see Figure 5—figure supplement 1A). The tumor cells are *Ddr2^+/+^*. In addition, we have now added an orthotopic transplant model in which *Ddr2^+/+^* tumor cells are implanted into the breast of syngeneic *Ddr2^+/+^* or *Ddr2^-/-^* hosts (New Figure 1C). We could do this now as we have backcrossed the *Ddr2^-/-^* allele onto a Balb/C background (>6 generations). In agreement with the spontaneous model we see a significant decrease in the number of lung metastases in *Ddr2^-/-^* transplanted mice. These 2 experiments indicate that the action of DDR2 in the primary tumor stroma, particularly in CAFs, impact lung metastases.

Perhaps, the reviewer wanted to see a transplant experiment performed in FSP1-Cre; Ddr2^fl/fl^ mice? Due to the 2-month window for a response we did not have a large enough colony FSP1-Cre; *Ddr2^-/-^* mice colony on a FVB/N background to perform a syngeneic transplant metastasis model using primary WT PyMT tumor cells (FVB/n), as we did in Corsa et al., 2016. Moreover, the FSP1-Cre; Ddr2^fl/fl^ alleles have not been backcrossed onto a pure Balb/C background, which precludes performing a 4T1 syngeneic transplant metastasis experiment.

With respect to the tail vein injection experiment presented, we completely agree with the reviewer – this is not a metastasis assay. It is a lung colonization assay. In other words, is the action of *Ddr2* in lung parenchymal cells (metastatic site) required for tumor cells to extravasate from the bloodstream, survive and grow in the lungs. This was done to test whether deletion of *Ddr2* in the lung stromal cells affected lung colonization by WT DDR2 tumor cells, as opposed to the action of DDR2 in the stromal cells of the primary tumor (assayed in the orthotopic transplant experiment Figure 1C). Throughout the revised text we have been careful to assure that we refer to this assay only as a lung colonization assay, not a metastasis assay.

With respect to the comment “Moreover, there is very little mechanistic work connecting the biology of the primary tumor with development of lung metastases” the current paper focuses exclusively on changes in the primary tumor matrix (collagen fiber architecture and physical properties) as a result of deletion of *Ddr2* in CAFs. Mechanistically, we believe we show in the current manuscript that the action of *Ddr2* in primary breast tumor CAFs is critical for the generation of an invasive permissive microenvironment – altered collagen fiber organization (SHG, FIB-SEM) leading to decreased primary tumor stiffness particularly at the tumor stromal boundary (AFM). This appears to be the result of *Ddr2* regulating full activation of collagen-binding Integrins through regulation of inside-out signaling regulating Talin1 activation/recruitment to Integrins. As to the action of DDR2 in primary tumor cells, our original publication in Nature Cell Biology 2012 showed that DDR2 activation by collagen stabilized SNAIL1 to sustain an EMT or mesenchymal phenotype and thus tumor cell invasion and migration through the collagen-rich ECM. Likely, DDR2 regulation of collagen binding Integrins (described in the current manuscript) in tumor cells also contributes to this defect in tumor cell invasion and migration (see Figure 7).

2) In the text, the authors contrast a published role for DDR2 in breast cancer lung metastasis in the presence of a primary tumor only (reciprocal orthotopic syngeneic breast tumor transplant experiments, Corsa CA 2016) with reciprocal tail vein injection in Ddr2^+/+^ vs. Ddr2^-/-^ host mice in which metastasis was unaffected by DDR2. Although these orthotopic data are previously published, they are foundational to the authors' overall claim regarding lung metastasis. The authors should repeat the orthotopic experiment and present this alongside the data in Figure 1.

We have now done this, and it is presented as new Figure 1B and C. Rather than repeat the same experiment as in Corsa et al., we performed a syngeneic orthotopic transplant metastasis assay using mouse 4T1 breast tumor cells. These are TNBC-like cells, that are highly invasive and metastatic. But they are derived from Balb/C mice. Fortunately, we had been backcrossing the *Ddr2^-/-^* allele onto a pure Balb/C background and there now >6 generations. This allowed us to perform the experiment in the time (2 month) alloted. As shown in new Figure 1B and C, there is significantly decreased lung metastases in *Ddr2^-/-^* mice transplanted with *Ddr2^+/+^* 4T1 breast tumor cells, compared to *Ddr2^+/+^* mice transplanted with *Ddr2^+/+^* 4T1 tumor cells.

3) Figure 3—figure supplement 1. Does Kindlin-2 remain at the focal adhesion sites in the shDDR2 cells? A co-localization of Kindlin-2 with total β1 integrin would help address this.

This was done. As observed for Talin1 and total β1-Integrin in *Ddr2*-depleted cells (Figure 3B), Kindlin2 and total β1-Integrin (Figure 3—figure supplement 1C and D) co-localized at focal adhesion sites in cellular protrusions in *Ddr2*-depleted cells, as determined by super-resolution confocal microscopy. This was in contrast to Talin1-active β1-Integrin (Figure 3C) and Kindlin2-active β1-Integrin (Figure 3—figure supplement 1A and B) which did not co-localize.

4) The images in Figure 7 from the tumor organoids are difficult to interpret. It seems that quantification was carried out in 3D z-stacks. The authors state that the active β1 integrin signal was reduced in CAFs but not in tumor cells in the Ddr2^-/-^ organoids. It is difficult to see how this assumption can be made based on the type of images shown. Furthermore, the quantification represents total signal throughout the organoid.

To address this we have quantified active β1-Integrin staining in CAFs (FAP+ cells) and non-FAP staining tumor cells in primary tumor organoids. To do so, imaging was performed via confocal microscopy (Zeiss, 63X). Analysis was performed using FIJI to quantify fluorescence (corrected total cell fluorescence (CTCF) = Integrated Density – (Area of selected cell X Mean fluorescence of background readings)) in the entire tumor organoid as well as in FAP positive cells only. This has been added to Figure 7 as panel E. Results showed that in CAFs (defined as FAP+ cells) in both FSP1-Cre deleted *Ddr2* tumor organoids and ubiquitous *Ddr2^-/-^* tumor organoids, active β1-Integrin staining was dramatically and significantly reduced.